# The serine protease hepsin mediates urinary secretion and polymerisation of Zona Pellucida domain protein uromodulin

Martina Brunati[1], Simone Perucca[1], Ling Han[2], Angela Cattaneo[3,4], Francesco Consolato[1], Annapaola Andolfo[4], Céline Schaeffer[1], Eric Olinger[5], Jianhao Peng[6], Sara Santambrogio[1], Romain Perrier[7], Shuo Li[6], Marcel Bokhove[2], Angela Bachi[3,4], Edith Hummler[7], Olivier Devuyst[5], Qingyu Wu[6], Luca Jovine[2], Luca Rampoldi[1]*

[1]Division of Genetics and Cell Biology, San Raffaele Scientific Institute, Milan, Italy; [2]Department of Biosciences and Nutrition & Center for Innovative Medicine, Karolinska Institutet, Huddinge, Sweden; [3]Functional Proteomics, FIRC Institute of Molecular Oncology, Milan, Italy; [4]Protein Microsequencing Facility, San Raffaele Scientific Institute, Milan, Italy; [5]Institute of Physiology, Zurich Center for Integrative Human Physiology, University of Zurich, Zurich, Switzerland; [6]Department of Molecular Cardiology, Lerner Research Institute, Cleveland, United States; [7]Department of Pharmacology and Toxicology, University of Lausanne, Lausanne, Switzerland

*For correspondence: rampoldi. luca@hsr.it

Competing interests: The authors declare that no competing interests exist.

**Abstract** Uromodulin is the most abundant protein in the urine. It is exclusively produced by renal epithelial cells and it plays key roles in kidney function and disease. Uromodulin mainly exerts its function as an extracellular matrix whose assembly depends on a conserved, specific proteolytic cleavage leading to conformational activation of a Zona Pellucida (ZP) polymerisation domain. Through a comprehensive approach, including extensive characterisation of uromodulin processing in cellular models and in specific knock-out mice, we demonstrate that the membrane-bound serine protease hepsin is the enzyme responsible for the physiological cleavage of uromodulin. Our findings define a key aspect of uromodulin biology and identify the first *in vivo* substrate of hepsin. The identification of hepsin as the first protease involved in the release of a ZP domain protein is likely relevant for other members of this protein family, including several extracellular proteins, as egg coat proteins and inner ear tectorins.

## Introduction

Uromodulin, also known as Tamm-Horsfall protein, is a 105 kDa glycoprotein exclusively expressed in the kidney by epithelial cells lining the thick ascending limb (TAL) of Henle's loop. It is a glycosylphosphatidylinositol (GPI)-anchored protein mainly localised at the apical plasma membrane of TAL epithelial cells. Uromodulin is secreted into the renal tubule through a conserved proteolytic cleavage (*Santambrogio et al., 2008*). In the urine it constitutes the most abundant protein under physiological conditions (*Serafini-Cessi et al., 2003*) and it is mainly found as high molecular-weight polymers. Uromodulin polymerisation is mediated by its Zona Pellucida (ZP) domain, a module present in several proteins that assemble into extracellular polymers, such as components of the egg

**eLife digest** Several proteins in humans and other animals contain a region called a 'zona pellucida domain'. This domain enables these proteins to associate with each other and form long filaments. Uromodulin is one such protein that was first identified more than fifty years ago. This protein is known to play a role in human diseases such as hypertension and kidney failure, but uromodulin's biological purpose still remains elusive.

Uromodulin is only made in the kidney and it is the most abundant protein in the urine of healthy individuals. Uromodulin also contains a so-called 'external hydrophobic patch' that must be removed before the zona pellucida domain can start to form filaments. This hydrophobic patch is removed when uromodulin is cut by an unknown enzyme; this cutting releases the rest of the uromodulin protein from the surface of the cells that line the kidney into the urine.

Brunati et al. have now tested a panel of candidate enzymes and identified that one called hepsin is able to cut uromodulin. Hepsin is embedded in the cell membrane of the cells that line the kidney. When the level of hepsin was artificially reduced in cells grown in the laboratory, uromodulin remained anchored to the cell surface, its processing was altered and it did not form filaments.

Brunati et al. next analysed mice in which the gene encoding hepsin had been deleted. While these animals did not have any major defects in their internal organs, they had much lower levels of uromodulin in their urine. Furthermore, this residual urinary protein was not cut properly and it did not assemble into filaments. Thus, these findings reveal that hepsin is the enzyme that is responsible for releasing uromodulin in the urine. This discovery could be exploited to alter the levels of uromodulin release, and further studies using mice lacking hepsin may also help to understand uromodulin's biological role. Finally, it will be important to understand if hepsin, or a similar enzyme, is also responsible for the release of other proteins containing the zona pellucida domain.

coat (including sperm receptors ZP3 and ZP2) and of the inner ear tectorial membrane (*Jovine et al., 2005*). Specific cleavage is necessary for protein assembly into polymers, as it releases an inhibitory hydrophobic motif (external hydrophobic patch, EHP) distal to the ZP domain that prevents polymerisation (*Jovine et al., 2004*; *Schaeffer et al., 2009*; *Han et al., 2010*).

The biological function of uromodulin is complex and only partly understood. Studies in $Umod^{-/-}$ mice demonstrated that it protects against urinary tract infections (*Bates et al., 2004*) and renal stone formation (*Liu et al., 2010*). Uromodulin has also been shown to modulate the activity of innate immunity cells via Toll-like receptor 4 signalling (*Säemann et al., 2005*) or via activation of the inflammasome (*Darisipudi et al., 2012*) and to regulate granulopoiesis and systemic neutrophil homeostasis (*Micanovic et al., 2015*). Finally, it is involved in the regulation of NaCl transport in the TAL, based on its role in promoting surface expression of the renal outer medullary potassium channel (ROMK2) (*Renigunta et al., 2011*) and of the sodium-potassium-chloride co-transporter (NKCC2) (*Mutig et al., 2011*).

Recent data showed that uromodulin plays an important role in chronic diseases of the kidney (*Rampoldi et al., 2011*). Mutations in the uromodulin gene (*UMOD*) cause an autosomal dominant tubulo-interstitial kidney disease (*Hart, 2002*). In addition, genome-wide association studies demonstrated that common variants in the *UMOD* gene are associated with increased risk for chronic kidney disease (CKD) and hypertension (*Köttgen et al., 2009*; *Padmanabhan et al., 2010*). Such an effect is due to higher *UMOD* expression driven by the presence of risk alleles in its gene promoter (*Trudu et al., 2013*).

Given the importance of polymerisation for uromodulin activity and the fact that this process depends on a specific protein cleavage, in this work we aimed at identifying the protease responsible for such cleavage and urinary secretion.

# Results

## Uromodulin cleavage and polymerisation in MDCK cells is mediated by a serine protease

As for other ZP proteins, uromodulin cleavage at a specific site in the protein C-terminus releases the interaction between two hydrophobic motifs (internal hydrophobic patch, IHP; external hydrophobic patch, EHP) (*Figure 1A*), leading to conformational activation of the ZP domain and protein polymerisation (*Jovine et al., 2004*; *Schaeffer et al., 2009*; *Han et al., 2010*).

To understand the nature of such cleavage, we took advantage of a cellular system, Madin-Darby Canine Kidney (MDCK) cells, where transfected human uromodulin assembles extracellularly in filamentous polymers (*Figure 1B,C*) that are indistinguishable from the urinary ones (*Jovine et al., 2002*). In these cells, uromodulin is secreted as two isoforms that can be separated on gel electrophoresis after enzymatic removal of protein N-glycans at about 72 and 77 kDa (*Figure 1D*). Only the shorter isoform assembles into polymers, since it is generated by a cleavage that releases the inhibitory EHP motif, while the longer one is generated by a more distal cleavage and still retains the EHP (*Schaeffer et al., 2009*). The short uromodulin isoform released by MDCK cells corresponds to the one found in the urine, as it shares the same molecular weight (*Figure 1D*) and the same

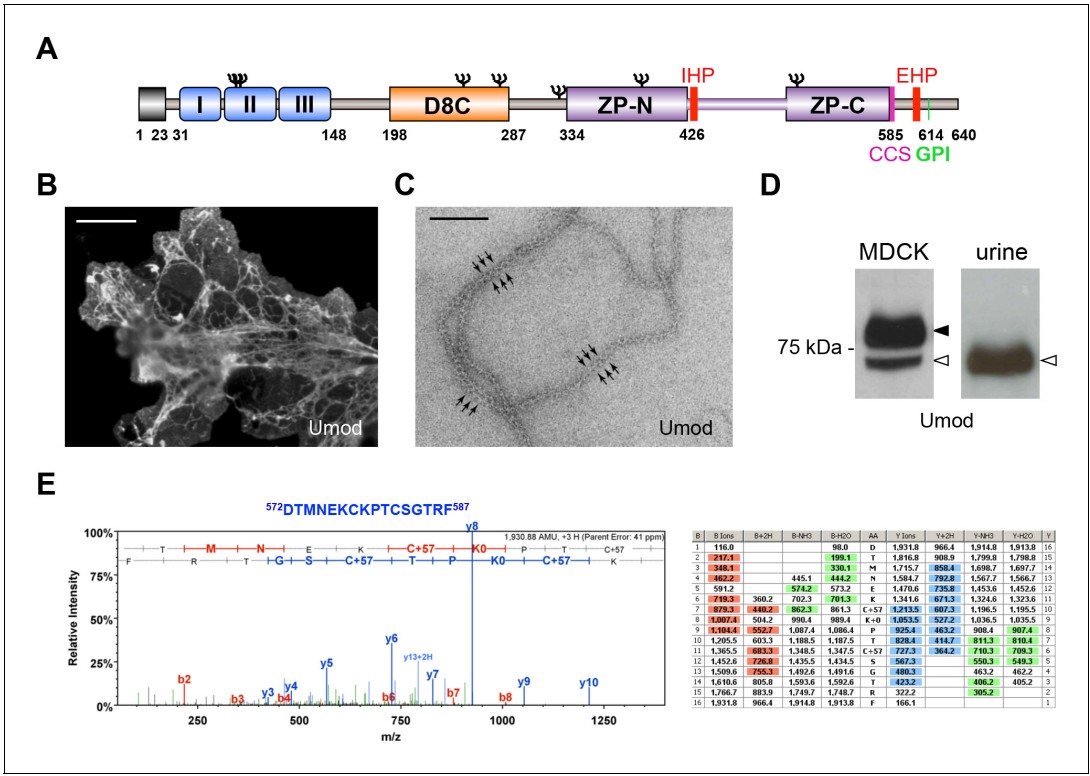

**Figure 1.** MDCK cells as a model to study physiological uromodulin shedding. (**A**) Schematic representation of human uromodulin domain structure containing a leader peptide (predicted to be cleaved at residue 23), three EGF-like domains, a central domain with 8 conserved cysteines (D8C), a bipartite Zona Pellucida (ZP) domain (ZP-N/ZP-C) and a glycosylphosphatidylinositol (GPI)-anchoring site (predicted at position 614). Internal (IHP) and External (EHP) Hydrophobic Patches (*Jovine et al., 2004*; *Schaeffer et al., 2009*), Consensus Cleavage Site (CCS) and seven N-glycosylation sites (Ψ) are also indicated. (**B**) Immunofluorescence analysis of non-permeabilised MDCK cells expressing uromodulin. Polymers formed by the protein are clearly detected on the cell surface. Scale bar, 50 μm. (**C**) Electron microscopy analysis of uromodulin polymers purified from the medium of MDCK cells. The arrows indicate the typical protrusions of uromodulin filaments spaced about 130 Å. Scale bar, 100 nm. (**D**) Representative Western blot analysis of N-deglycosylated uromodulin secreted by transfected MDCK cells or purified from urine. A single isoform is clearly seen in the urinary sample. An isoform with similar molecular weight is released by MDCK cells (white arrowhead), which also secrete a longer and more abundant one (black arrowhead). (**E**) Representative tandem mass-spectrometry (MS/MS) spectrum confirming the identity of the C-terminal peptide [572]DTMNEKCKPTCSGTRF[587] of the short uromodulin isoform released by MDCK cells and table of fragmented ions. The C-terminal residue F587 is identical to the one that we mapped in human urinary protein (*Santambrogio et al., 2008*).

C-terminal residue (F587 [*Santambrogio et al., 2008*]) (*Figure 1E*), demonstrating that uromodulin undergoes physiological cleavage in these cells.

As mapping of the C-terminus of the short uromodulin isoform suggests proteolytic cleavage, we first treated uromodulin-expressing MDCK cells with a protease inhibitor cocktail (PIC). This treatment led to significant reduction of uromodulin polymerisation on the surface of the cells (*Figure 2A*) that is not due to any alteration of protein expression or intracellular distribution (*Figure 2—figure supplement 1*), confirming that the polymerisation-competent uromodulin isoform is generated through the action of a protease. We then treated MDCK cells with the single PIC components, each targeting a specific catalytic type of protease, and observed that only inhibitors of serine proteases significantly reduced uromodulin polymerisation (*Figure 2A*). Consistently, the expression of the Kunitz-type serine protease inhibitor Hepatocyte growth factor Activator Inhibitor-1 (HAI-1) (*Shimomura et al., 1997*) essentially abolished uromodulin polymerisation on the surface of MDCK cells and reduced the secretion of the shorter uromodulin isoform (*Figure 2—figure supplement 2*). To explore the role of membrane-association for uromodulin cleavage, we expressed a soluble isoform of uromodulin, generated by truncation at the GPI-anchoring site (S614X), that is efficiently trafficked and secreted in MDCK cells (*Schaeffer et al., 2009*). Interestingly, this isoform was neither processed to the shorter variant nor it formed polymers (*Figure 2B,C*), indicating that, as in the case of ZP3 (*Jovine et al., 2002*), membrane association is necessary to allow uromodulin physiological cleavage and polymerisation. From these results we conclude that physiological cleavage of uromodulin depends on a serine protease, and that this enzyme is likely membrane associated.

## Identification of hepsin as the protease cleaving uromodulin *in vitro*

We moved further starting from the interesting observation that uromodulin expressed in a different kidney-derived cell line, Human Embryonic Kidney 293 (HEK293), did not form polymers on the cell surface (*Figure 3A*). Accordingly, only the longer polymerisation-incompetent uromodulin isoform was released by these cells (*Figure 3B*). We hypothesised that this was due to lack of expression of the proper serine protease responsible for physiological uromodulin cleavage. Under this assumption, we compared the expression profiles available in Gene Expression Omnibus of known serine proteases (about 220 enzymes) between MDCK and HEK293 cells, to identify those exclusively expressed by the former. Among the resulting 23 candidates we then selected only membrane-bound enzymes that were found to be expressed in available TAL segment transcriptomes (*Yu et al., 2009*; *Cheval et al., 2011*). This strategy led to the identification of only four candidate enzymes: prostasin (Prss8), hepsin/TMPRSS1 (Hpn), mucin 1 (Muc1) and dipeptidyl-peptidase IV (Dpp4) (*Figure 3C*). The last two enzymes were excluded on the basis of their catalytic activity: Dpp4 sequentially removes N-terminal dipeptides from polypeptides, cleaving after a proline residue (*Mentlein, 1999*), whereas Muc1 contains a self-cleaving module rather than a classical peptidase S1 domain (*Levitin et al., 2005*). Validation of transcriptome data confirmed differential expression of the two candidate proteases, hepsin and prostasin, in MDCK and HEK293 cells (*Figure 3—figure supplement 1A*). Interestingly, both proteases are inhibited by the Kunitz-type serine protease inhibitor HAI-1 (*Fan et al., 2005*; *Herter et al., 2005*), in line with our results in MDCK cells (*Figure 2—figure supplement 2*).

We validated our hypothesis by expressing either hepsin or prostasin in HEK293 cells. This induced short cleavage as well as polymerisation of uromodulin, an effect that was dependent on the catalytic activity of the enzymes, as it was abolished when catalytically inactive isoforms of the proteases were expressed (*Figure 3D,E*; *Figure 3—figure supplement 1B*). Moreover, this effect is likely direct, as both enzymes could be specifically co-immunoprecipitated with uromodulin from the lysates of HEK293 cells (*Figure 3F*). Based on this observation, we assessed whether uromodulin can be directly cleaved by hepsin and prostasin *in vitro*. To do so, we carried out cleavage assays by employing a truncated recombinant form of uromodulin that corresponds to the elastase-resistant fragment of uromodulin from human urine (efUmod, lacking residues 27–294) (*Jovine et al., 2002*). We focussed on this fragment as it primarily consists of the ZP polymerisation domain (*Figure 4A*). Also, deletion of the N-terminal part of uromodulin greatly improved its purification efficiency (data not shown), probably because it lacks several N-glycosylation sites. Importantly, this isoform undergoes proper proteolytic cleavage and forms polymers in MDCK cells (*Figure 4B*). When incubated with purified efUmod, both hepsin and prostasin induced a mass shift of efUmod as well as loss of its C-terminal histidine tag (*Figure 4C*). These effects were abolished by mutagenesis of the

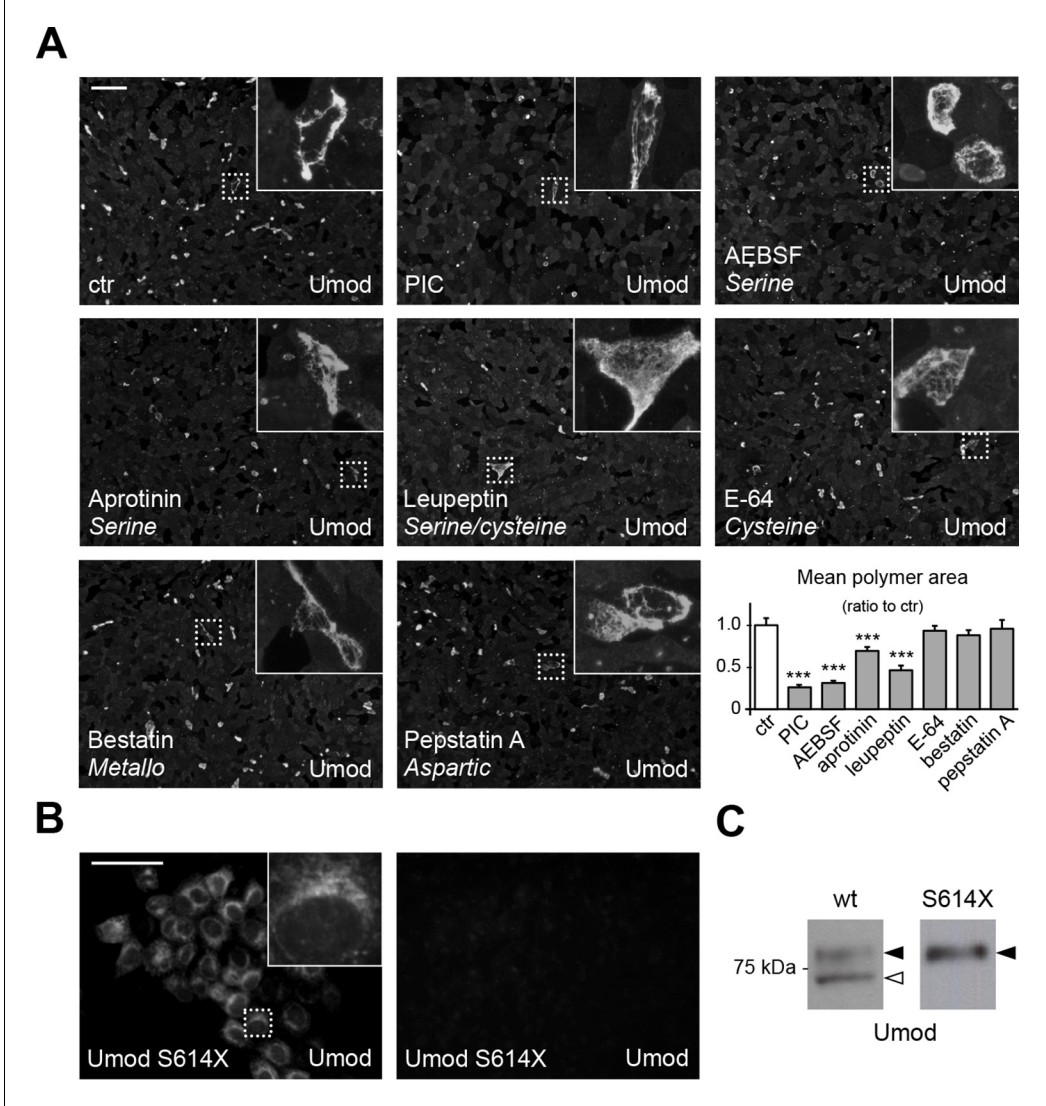

**Figure 2.** A serine protease is responsible for the release of polymerisation-competent uromodulin. (**A**) Immunofluorescence analysis showing uromodulin on the surface of MDCK cells treated with vehicle (DMSO) (ctr), protease inhibitor cocktail (PIC) or single PIC components, as indicated. Scale bar, 50 μm. Quantification of the average surface of uromodulin polymers shows that only treatment with PIC or with specific serine protease inhibitors (AEBSF, aprotinin and leupeptin) significantly reduces uromodulin polymerisation on the surface of MDCK cells. Bars indicate average ± s.e.m. ***$p < 0.001$ (Mann-Whitney test). The graph represents mean ratios of 3 independent experiments (*Figure 2—source data 1*). (**B**) Immunofluorescence analysis of permeabilised (left) or non-permeabilised (right) MDCK cells expressing soluble uromodulin mutant S614X. Uromodulin polymerisation is abolished by preventing its association to the membrane. Scale bar, 50 μm. (**C**) Representative Western blot analysis of N-deglycosylated uromodulin secreted by MDCK cells expressing wild-type or soluble (S614X) uromodulin. Lack of plasma membrane anchoring does not affect uromodulin secretion but it abolishes its cleavage at the physiological site (white arrowhead).

The following source data and figure supplements are available for figure 2:

**Source data 1.** Quantification of the area of uromodulin polymers on the surface of MDCK cells after protease inhibitor treatment (*Figure 2A*).

**Source data 2.** Short cleavage inhibition by HAI-1 expression (*Figure 2—figure supplement 2C*).

**Figure supplement 1.** PIC treatment does not affect uromodulin intracellular distribution and expression.

**Figure supplement 2.** Expression of the serine protease inhibitor HAI-1 effectively reduces uromodulin cleavage at the urinary site.

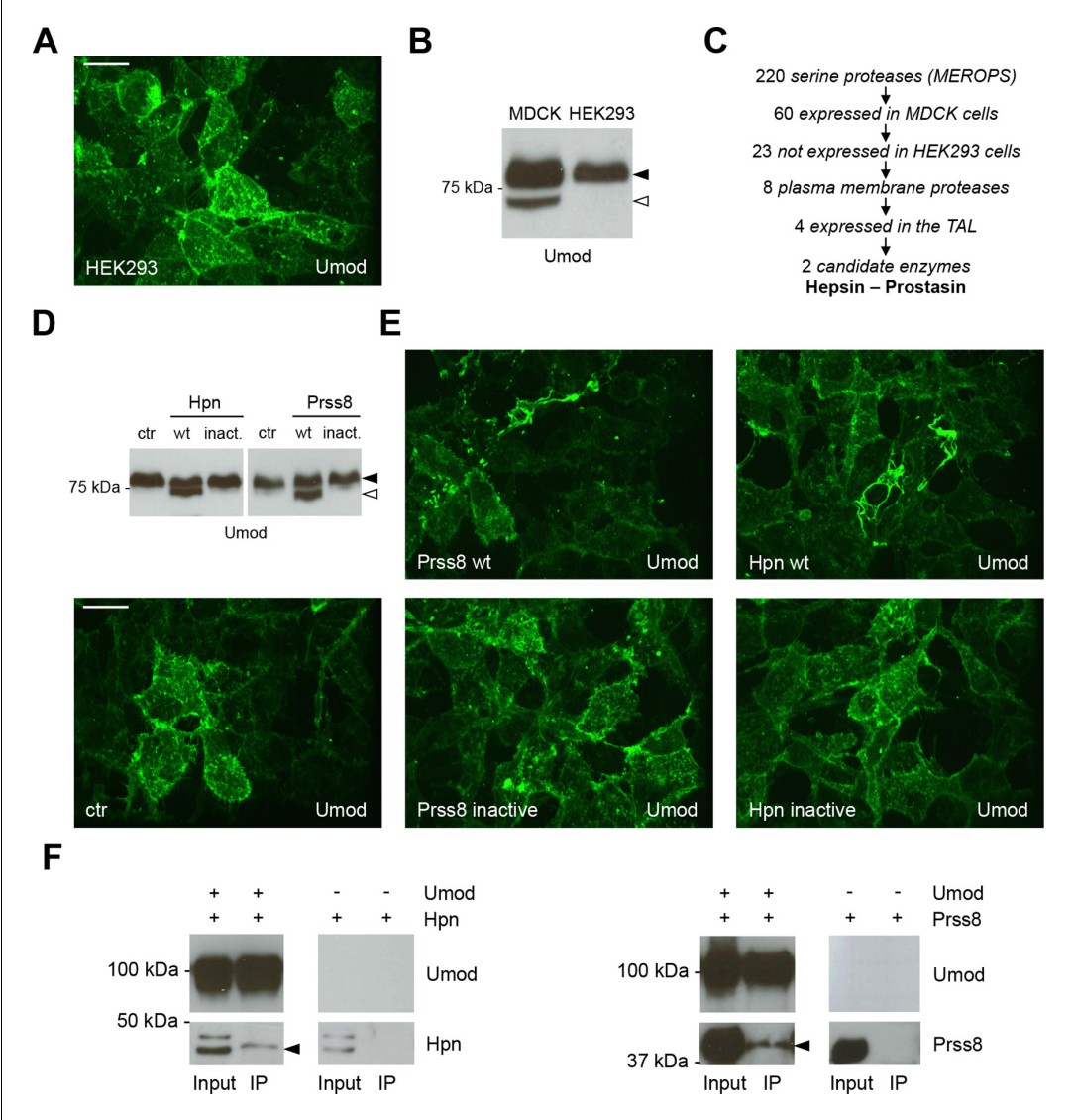

**Figure 3.** Hepsin and prostasin interact with uromodulin to induce its cleavage and polymerisation. (**A**) Immunofluorescence analysis (extended focus image) showing uromodulin on the surface of HEK293 cells. The protein is not assembled into polymers when expressed in this cellular system. Scale bar, 21 µm. (**B**) Representative Western blot analysis of N-deglycosylated uromodulin secreted by MDCK and HEK293 cells. HEK293 cells only secrete the longer polymerisation-incompetent uromodulin isoform (black arrowhead), while MDCK cells also secrete the shorter one (white arrowhead). (**C**) Schematic representation of the selection process employed to identify candidate enzymes for the secretion of the short uromodulin isoform. Only membrane-bound serine proteases specifically expressed by MDCK cells and by the TAL segment of the nephron, but not by HEK293 cells, were selected. (**D**) Representative Western blot analysis of N-deglycosylated uromodulin secreted by HEK293 cells expressing wild-type or catalytically inactive human hepsin or prostasin, as indicated. Only wild-type proteases promote the secretion of the short uromodulin isoform by these cells (white arrowhead). (**E**) Immunofluorescence analysis (extended focus image) showing uromodulin on the surface of HEK293 cells expressing wild-type or catalytically inactive hepsin or prostasin, as indicated. Uromodulin polymerisation is induced only when wild-type proteases are expressed. Scale bar, 21 µm. (**F**) Representative Western blot analysis of uromodulin immunoprecipitation (upper panels) from lysates of HEK293 cells expressing hepsin or prostasin, as indicated. Both enzymes are co-immunoprecipitated when uromodulin is co-expressed in HEK293 cells (lower panels). The arrowheads point at hepsin and prostasin specific bands.

The following figure supplement is available for figure 3:

**Figure supplement 1.** Hepsin and prostasin expression in MDCK and HEK293 cells.

consensus cleavage site (efUmod YAla, carrying the mutation $^{586}$RFRS$^{589}$ > $^{586}$AYAA$^{589}$). Taken together, these findings indicate that hepsin and prostasin physically interact with uromodulin and

cleave it at the physiological site. Interestingly, hepsin was more efficient than prostasin, since it completely digested the substrate despite being used at lower concentration (see figure legend).

The two candidate proteases could not be discriminated on the basis of their subcellular localisation in MDCK cells. Indeed, while endogenous proteases could not be detected, possibly because of low expression levels or low affinity for the employed antibodies (which were raised against their human counterparts), transfected hepsin and prostasin both co-localised with uromodulin on the apical membrane of polarised MDCK cells (*Figure 5A*). Hence, we designed short hairpin RNAs (shRNAs) to specifically silence either protease in this cellular system. We successfully obtained comparable expression knock-down for both enzymes (*Figure 5B*), but only hepsin silencing led to a significant reduction of uromodulin polymerisation on the cell surface (*Figure 5C*). These results demonstrate that hepsin is the enzyme responsible for uromodulin cleavage at the urinary site in MDCK cells.

## Hepsin mediates physiological cleavage and polymerisation of uromodulin *in vivo*

To understand the physiological relevance of our findings, we then investigated the expression of endogenous hepsin in mouse kidneys. Remarkably, by microdissecting mouse nephron segments, we detected strong hepsin expression in the TAL (*Figure 6A*). Moreover, hepsin co-localises with uromodulin on the apical membrane of TAL epithelial cells, suggesting a functional interaction between the two proteins (*Figure 6B*). To assess the role of hepsin in uromodulin processing *in vivo*, we characterised uromodulin urinary secretion in a previously generated $Hpn^{-/-}$ mouse model, which does not show any gross phenotype or structural abnormality in major organs (*Wu et al., 1998*). Notably, urinary uromodulin levels were significantly reduced in $Hpn^{-/-}$ mice (*Figure 6C*). This was accompanied by marked accumulation of the mature, fully glycosylated form of uromodulin in kidney lysates (*Figure 6D*). On the other hand, no difference was observed in the cellular distribution of uromodulin in TAL cells, showing typical apical membrane enrichment (*Figure 6E*), and no change in *Umod* gene expression was detected (*Figure 6F*). These data exclude that reduced levels of urinary uromodulin in $Hpn^{-/-}$ mice could be due to changes in protein expression or transport to the apical plasma membrane, strongly suggesting defective protein secretion.

After deglycosylation, urinary uromodulin from $Hpn^{-/-}$ mice appeared as two isoforms: a short one, with similar electrophoretic mobility as in wild-type urines, and a longer one, normally undetectable in urines from wild-type mice (*Figure 7A*). Mass-spectrometry analysis on the short isoform showed that it lacks the physiological C-terminus (F588) and is produced by a more distal cleavage occurring after residue R607 (*Figure 7B–D*). The same analysis on the long isoform revealed an even more distal C-terminus, after residue K616 (*Figure 7—figure supplement 1*), consistent with its slower electrophoretic mobility. These data demonstrate that absence of hepsin abolishes uromodulin cleavage at the physiological site and induces its misprocessing. Interestingly, uromodulin from $Hpn^{-/-}$ urines was not sedimented in the pellet fraction of a biochemical polymerisation assay (*Jovine et al., 2002*) (*Figure 7E*), indicating that the residual urinary protein is not assembled into polymers. This is consistent with the observation that both urinary isoforms in $Hpn^{-/-}$ mice are misprocessed and still retain the polymerisation-inhibiting EHP motif (*Figure 7D*) (*Jovine et al., 2004*; *Schaeffer et al., 2009*).

A role for hepsin in the activation of prostasin, whose zymogen is unable to auto-activate, was demonstrated *in vitro* (*Chen et al., 2010*). To assess whether the results observed in $Hpn^{-/-}$ mice could be due to the lack of hepsin-mediated activation of prostasin, we first investigated the expression pattern of endogenous prostasin in mouse kidneys. Confirming data from available transcriptomes, we detected weak prostasin expression in TAL segments where it co-localises with uromodulin on the apical plasma membrane of epithelial cells (*Figure 7—figure supplement 2A,B*). We then generated an adult nephron-specific $Prss8^{-/-}$ mouse model and analysed urinary uromodulin. Secretion of urinary uromodulin in these mice was comparable to controls (*Figure 7—figure supplement 2C,D*). Moreover, mass-spectrometry analysis on the urinary protein confirmed that it shares the same C-terminal residue of control animals (*Figure 7—figure supplement 2E,F*), excluding a role for prostasin in uromodulin processing *in vivo*.

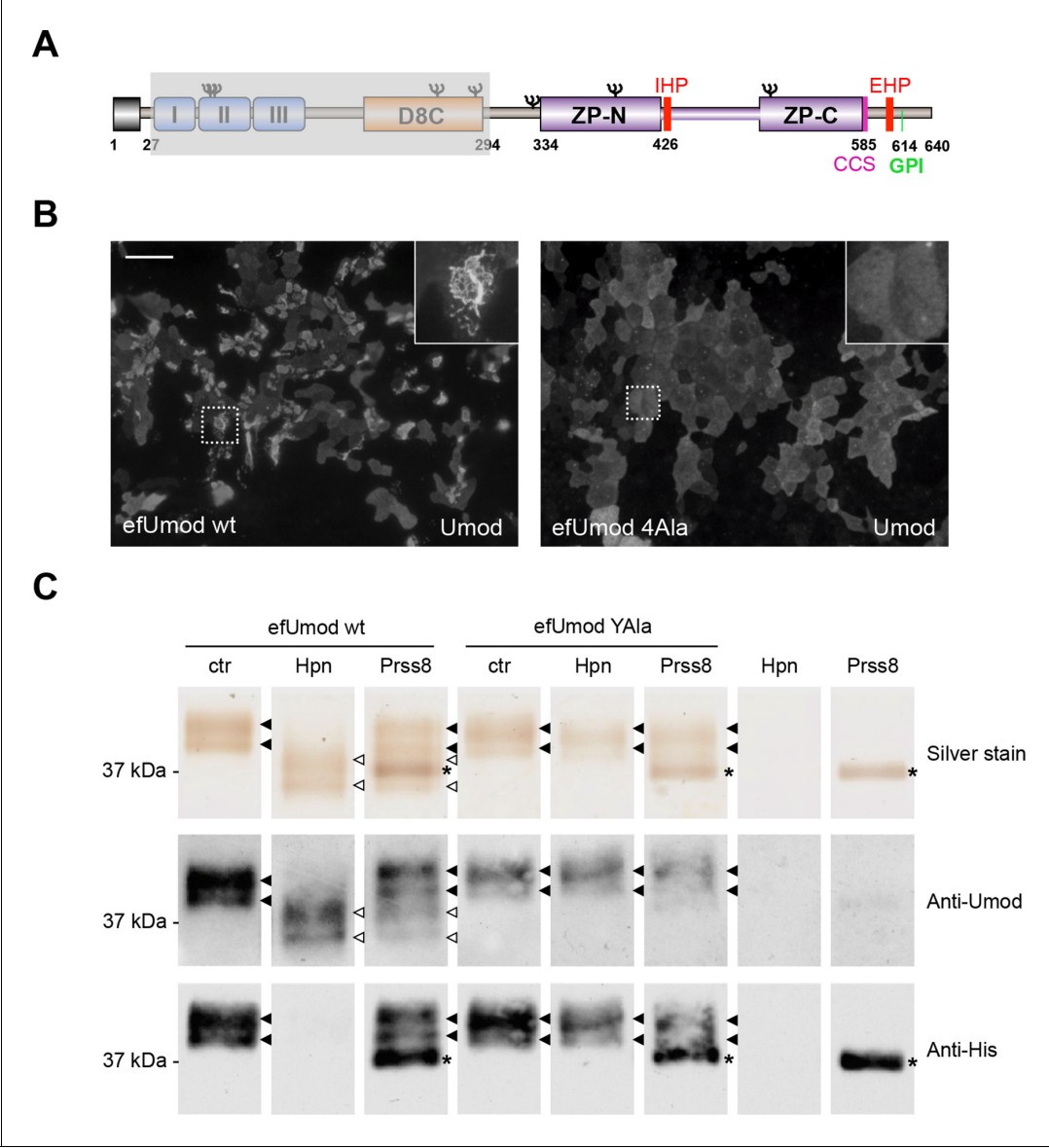

**Figure 4.** Hepsin and prostasin directly cleave uromodulin *in vitro*. (**A**) Schematic representation of human uromodulin domain structure as shown in *Figure 1A*. The region not included in recombinant efUmod is shadowed. (**B**) The deletion of the elastase-sensitive fragment of uromodulin does not affect protein polymerisation on the surface of MDCK cells, as shown by immunofluorescence analysis (efUmod wt). As for full-length uromodulin (*Schaeffer et al., 2009*), this process depends on correct protein cleavage at the physiological site, since it is abolished when the consensus cleavage site is mutated (efUmod 4Ala, carrying the mutation $^{586}$RFRS$^{589}$ > $^{586}$AAAA$^{589}$). Scale bar, 50 μm. (**C**) Purified efUmod, either wild-type (efUmod wt) or mutated at the consensus cleavage site (efUmod YAla, carrying the mutation $^{586}$RFRS$^{589}$ > $^{586}$AYAA$^{589}$), was incubated with recombinant prostasin or hepsin, as indicated. Both proteases decrease the mass of wild-type efUmod (white arrowheads in upper and middle panels) and cause the loss of its C-terminal His-tag (lower panel). Hepsin is more efficient than prostasin, as it drives complete digestion of the product, despite being used at 20x lower concentration (picomolar ratio between protease and efUmod was 1:100 for hepsin and 1:5 for prostasin, see lanes 7 and 8 for comparison). The asterisk indicates His-tagged prostasin.

## Discussion

Collectively, our results identify hepsin as the enzyme responsible for the physiological release of uromodulin in the urine. Hepsin, or TMPRSS1, is a member of type II transmembrane serine proteases. It is expressed in several tissues and it has been previously implicated in the processing of

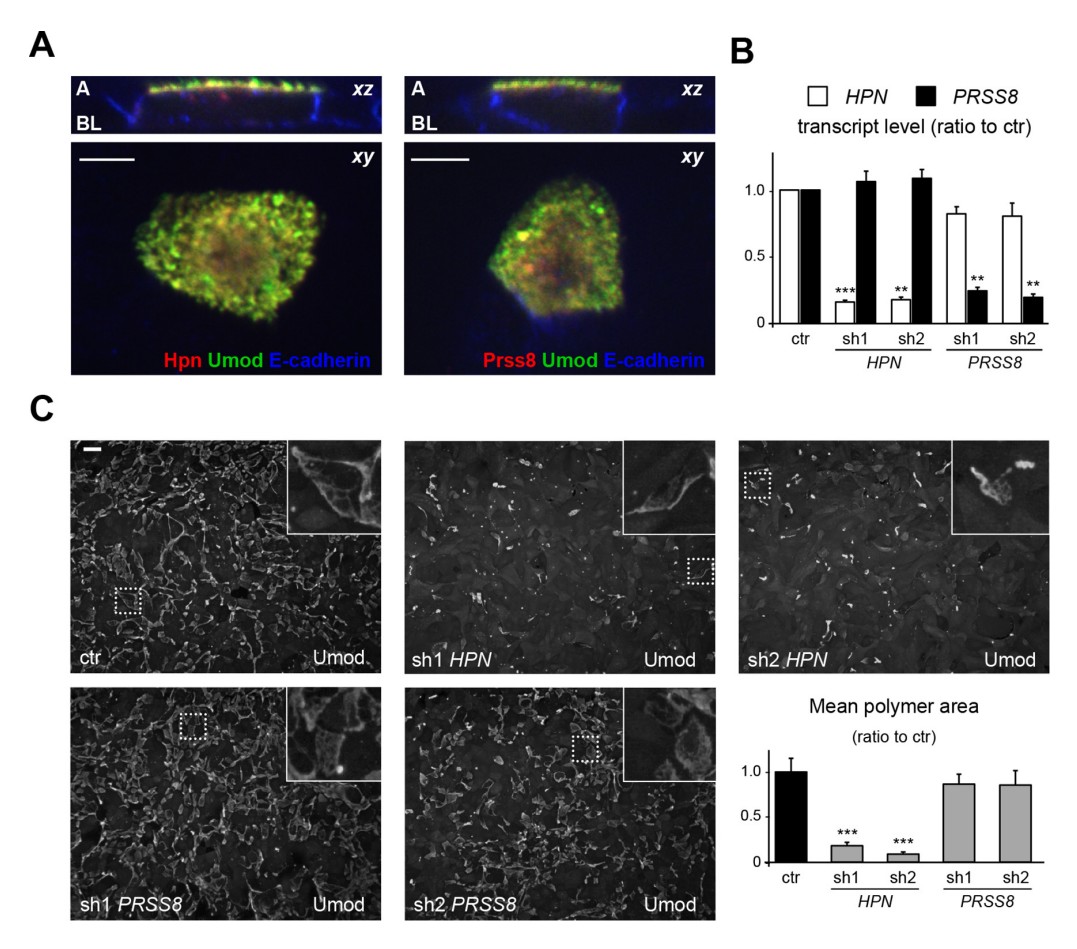

**Figure 5.** Hepsin is the protease mediating uromodulin polymerisation in MDCK cells. (A) Confocal immunofluorescence analysis showing uromodulin (green), hepsin or prostasin (red) and E-cadherin (blue) (basolateral membrane marker) in polarised MDCK cells, as indicated. Upper panels represent the reconstruction on the *xz* axis of merged *xy* scans, for which a representative image is shown in lower panels. Both serine proteases co-localise with uromodulin on the apical plasma membrane of polarised MDCK cells. *z* stacks = 0.3 μm. A: apical, BL: basolateral. Scale bars, 5 μm. (B) Transcript levels of *HPN* and *PRSS8*, as assessed by Real-Time qPCR in MDCK cells transfected with shRNA vectors, as indicated. Expression values (normalised to glyceraldehyde-3-phosphate dehydrogenase, GAPDH) are shown as relative to cells transfected with control vector. Expression of the proteases is specifically reduced in silenced cells. Bars indicate average ± s.e.m. **p<0.01, ***p<0.001 (Student's *t* test). The graph represents mean ratios of 3 independent experiments (*Figure 5—source data 1*). (C) Immunofluorescence analysis showing uromodulin on the surface of MDCK cells transfected with control vector or with shRNA vectors targeting hepsin or prostasin, as indicated. Scale bar, 50 μm. Quantification of the average surface of uromodulin polymers shows that silencing of hepsin, but not of prostastin, substantially reduces uromodulin polymerisation on the membrane of MDCK cells. Bars indicate average ± s.e.m. ***p<0.001 (Mann-Whitney test). The graph represents mean ratios of 3 independent experiments (*Figure 5—source data 2*).

The following source data is available for figure 5:

**Source data 1.** Transcript level of *HPN* and *PRSS8* in MDCK cells after shRNA transfection (*Figure 5B*).

**Source data 2.** Quantification of the area of uromodulin polymers on the surface of MDCK cells after shRNA transfection (*Figure 5C*).

different substrates; however, the physiological relevance of most of these observations remains to be established (*Chen et al., 2010*; *Hsu et al., 2012*; *Khandekar and Jagadeeswaran, 2014*; *Ganesan et al., 2011*). By identifying hepsin as the enzyme responsible for the release of uromodulin in the renal tubule, our results reveal an important biological function of this type II transmembrane serine protease in the renal epithelium. At the same time, they represent a major advancement in understanding the biology of uromodulin, opening new avenues of research on the regulation and possibly modulation of its secretion. This could have impact for diseases such as urolithiasis where

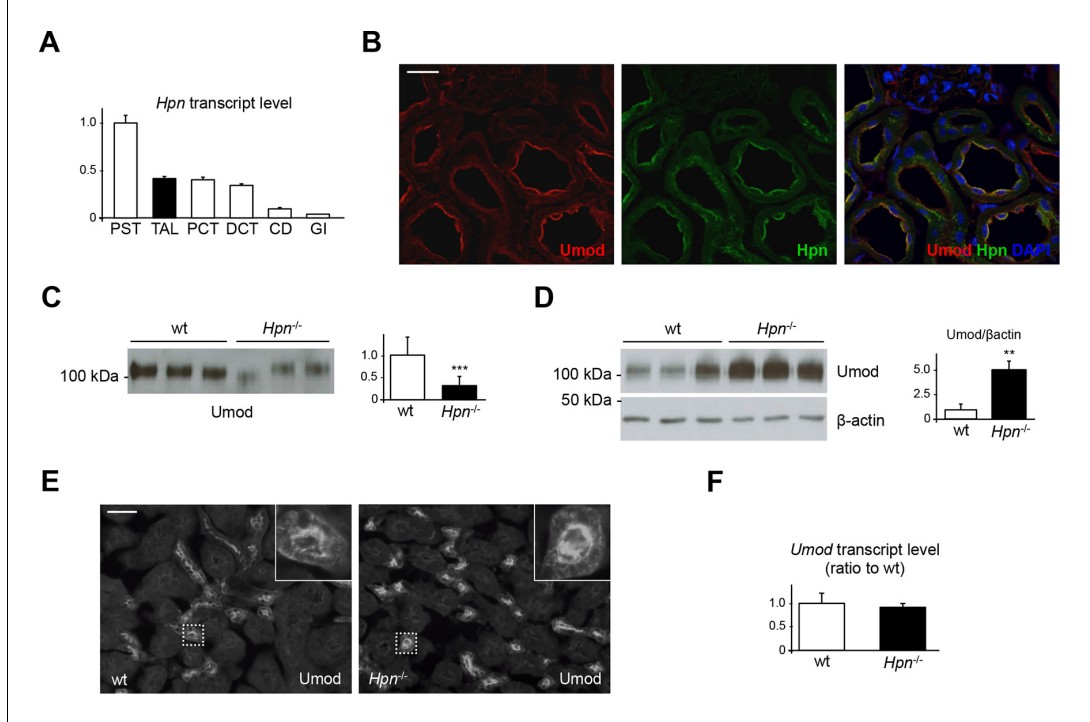

**Figure 6.** Defective uromodulin urinary secretion in mice lacking hepsin. (**A**) Transcript level of *Hpn*, as assessed by Real-Time qPCR on microdissected nephron segments (normalised to *Gapdh*). Expression of *Hpn* is detected in proximal straight tubules (PST), thick ascending limb (TAL), proximal convoluted tubules (PCT), distal convoluted tubules (DCT) and, to a lesser extent, in collecting ducts (CD). Minimal expression of the protease is detected in glomeruli (GI). Bars indicate average ± s.e.m. of 3 independent experiments (*Figure 6—source data 1*). (**B**) Immunofluorescence analysis of mouse kidney sections showing co-localisation of endogenous hepsin and uromodulin on the apical plasma membrane of TAL epithelial cells. Scale bar, 20 µm. (**C**) Representative Western blot analysis of urinary uromodulin secretion in $Hpn^{-/-}$ mice or control animals. Urinary protein loading was normalised to urinary creatinine concentration. Densitometric analysis shows reduced uromodulin urinary secretion in animals lacking hepsin (average ± s.d., n = 10/group, *Figure 6—source data 2*). ***p<0.001 (Student's *t* test). (**D**) Western blot analysis of uromodulin in kidney lysates of $Hpn^{-/-}$ mice or control animals. Beta-actin is shown as a loading control. Densitometric analysis shows accumulation of uromodulin in kidney lysates from $Hpn^{-/-}$ mice (average ± s.d., n = 3/group, *Figure 6—source data 3*). **p<0.01 (Student's *t* test). (**E**) Representative immunofluorescence analysis showing apical plasma membrane signal of uromodulin in kidney sections from $Hpn^{-/-}$ mice or control animals. Scale bar, 50 µm. (**F**) Transcript level of *Umod*, as assessed by Real-Time qPCR on total kidney extracts from $Hpn^{-/-}$ mice or control animals (normalised to *Hprt1*). Expression of *Umod* is comparable between the two groups of animals (n = 3/group, *Figure 6—source data 4*). Bars indicate average ± s.e.m. (Student's *t* test).

The following source data is available for figure 6:

**Source data 1.** Transcript level of *Hpn* in mouse microdissected nephron segments (*Figure 6A*).

**Source data 2.** Quantification of urinary uromodulin secretion in $Hpn^{-/-}$ and control mice (*Figure 6C*).

**Source data 3.** Quantification of uromodulin levels in kidney lysates of $Hpn^{-/-}$ and control mice (*Figure 6D*).

**Source data 4.** Transcript levels of *Umod* in $Hpn^{-/-}$ and control mice (*Figure 6F*).

the urinary levels of uromodulin may be directly correlated with its protective function. Also, it might be relevant for CKD and hypertension where we demonstrated that *UMOD* risk variants are associated with increased uromodulin expression and urinary levels (*Trudu et al., 2013*; *Olden et al., 2014*). Lack of hepsin does not affect baseline renal function of $Hpn^{-/-}$ mice (serum creatinine: wild-type 0.78 +/- 0.04 vs $Hpn^{-/-}$ 0.74 +/- 0.14 mg/dL, n = 5/group, p=0.59; blood urea nitrogen: wild-type 21.6 +/- 3.29 vs $Hpn^{-/-}$ 20.8 +/- 4.12 mg/dL, n= 5/group, p=0.76). In this light, further studies in $Hpn^{-/-}$ mice with altered uromodulin processing and secretion are warranted to investigate if the mice develop a phenotype upon additional challenges. The results may provide new insight into uromodulin cellular and extracellular functions.

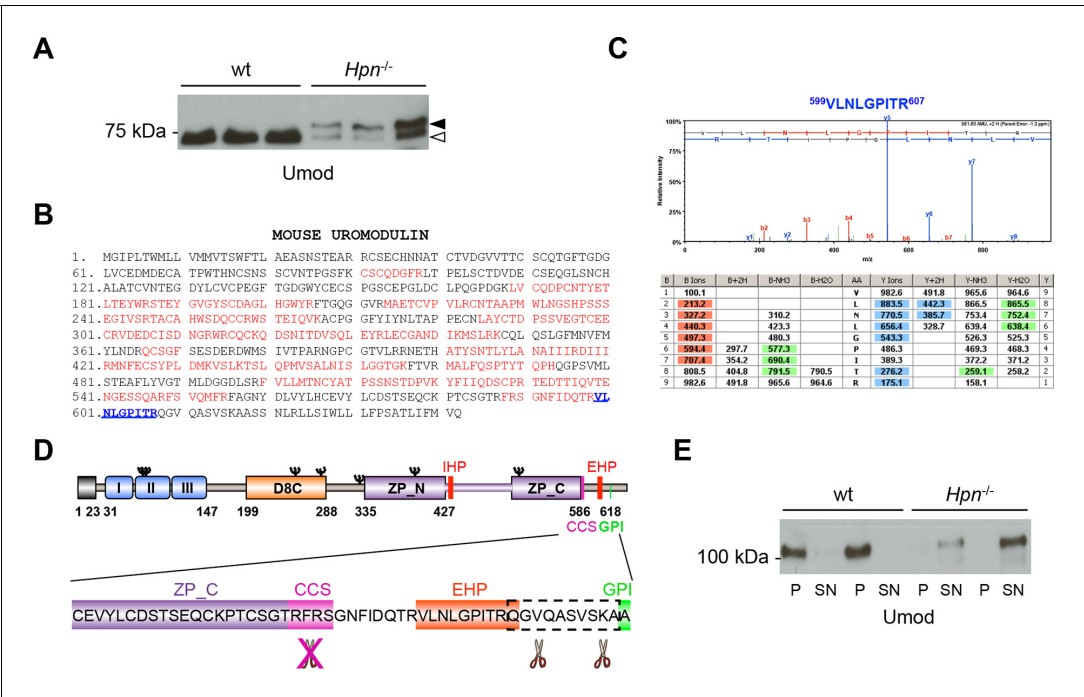

**Figure 7.** Absence of hepsin *in vivo* abolishes physiological cleavage and polymerisation of uromodulin. (**A**) Representative Western blot analysis of N-deglycosylated urinary uromodulin secreted by *Hpn*[-/-] mice or control animals. *Hpn*[-/-] mice show the presence of two uromodulin isoforms: a short one with similar electrophoretic mobility as in wild-type urines (white arrowhead), and a longer one that is absent in samples from wild-type mice (black arrowhead) (n = 6/group). (**B**) Mass spectrometry (MS) sequence coverage (52% over the entire protein) of trypsin-digested mouse uromodulin (short isoform) (UniProt accession Q91X17) purified from urine of *Hpn*[-/-] mice. Matching peptides are shown in red, while the C-terminal peptide is shown in blue. This peptide ends at R607, a distal C-terminal residue with respect to the one reported for mouse urinary uromodulin (F588 [*Santambrogio et al., 2008*]). (**C**) Representative tandem mass-spectrometry (MS/MS) spectrum, confirming the identity of the identified C-terminal peptide ([599]VLNLGPITR[607]) of the short uromodulin isoform released by *Hpn*[-/-] mice, and table of fragmented ions. (**D**) Schematic representation of mouse uromodulin domain structure as in *Figure 1A*. The blow-up shows that in the absence of hepsin, the cleavage generating the short uromodulin isoform is abolished and alternative ones at more C-terminal sites (distal to R607) take place. (**E**) Representative Western blot analysis of uromodulin in supernatant (SN) and pellet (P) fractions from a polymerisation assay performed on urinary samples from *Hpn*[-/-] mice or control animals. Urinary uromodulin from control animals is precipitated in the pellet fraction, reflecting full engagement in polymeric structures, while the one from *Hpn*[-/-] mice is only detected in the supernatant (n = 4/group).

The following source data and figure supplements are available for figure 7:

**Source data 1.** Transcript level of *Prss8* in mouse microdissected nephron segments (*Figure 7—figure supplement 2A*).

**Source data 2.** Quantification of urinary uromodulin secretion in *Prss8*[-/-] and control mice (*Figure 7—figure supplement 2C*).

**Figure supplement 1.** Urinary uromodulin misprocessing in *Hpn*[-/-] mice.

**Figure supplement 2.** Uromodulin secretion is not affected by lack of prostasin *in vivo*.

It is well recognised that proteolysis plays a central role in uromodulin polymerisation. This step leads to the release of an inhibitory intramolecular interaction that prevents premature protein polymerisation (*Jovine et al., 2004*; *Schaeffer et al., 2009*; *Han et al., 2010*). Considering the maturation steps of uromodulin along the secretory compartment, hepsin-mediated cleavage likely takes place at the plasma membrane where the two proteins co-localise, allowing the protein to traffic within the cell in a polymerisation-incompetent conformation, and to polymerise only once it is secreted (*Figure 8*). Interestingly, the C-terminus of human and mouse urinary uromodulin has been mapped to a phenylalanine residue (F587 and F588, respectively) (*Santambrogio et al., 2008*). Since *in vitro* studies showed that hepsin preferentially cleaves substrates after basic amino acids (*Herter et al., 2005*; *Béliveau et al., 2009*; *Owen et al., 2010*), this could result from cleavage at

the following arginine residue and subsequent C-terminal trimming. Post-cleavage trimming by carboxypeptidases is not uncommon and it has already been proposed for several secreted proteins, including urokinase (*Günzlerwa et al., 1982*), and ZP subunits (*Kubo et al., 1999*; *Boja et al., 2003*; *Darie et al., 2005*).

Through the identification of hepsin as the protease releasing uromodulin in the urine we discover the first protease involved in the physiological cleavage of a ZP domain protein, which is a pre-requisite for polymerisation of this type of molecules. Notably, $Hpn^{-/-}$ mice were reported to have profound hearing loss likely due to inner ear defects, including a deformed and enlarged tectorial membrane (*Guipponi et al., 2007*). This acellular structure, which overlays the organ of Corti, consists of filaments largely composed of ZP domain proteins α- and β-tectorin. The morphology of the tectorial membrane is similarly altered in mice carrying deafness-causing mutations in α-tectorin (*Legan et al., 2014*) that are likely affecting protein secretion (*Jovine et al., 2002*). These findings, along with data presented in this manuscript, suggest that inner ear tectorins are also physiological substrates of hepsin. Moreover, release of glycoprotein-2, a ZP domain protein highly homologous to uromodulin that is predominantly expressed in pancreas, is mediated by an apical protease sensitive to serine protease inhibitors (*Fritz and Lowe, 1996*), again suggesting the involvement of hepsin. In light of these observations and given the conserved function of the ZP domain and the high similarity of the consensus cleavage site of ZP domain proteins (*Jovine et al., 2004*), our results are likely to be relevant for other members of this protein family.

## Materials and methods

### Constructs

Uromodulin vectors expressing wild-type HA- or FLAG-tagged protein, soluble mutant S614X and consensus cleavage site mutant $^{586}$RFRS$^{589/586}$AAAA$^{589}$ (4Ala mutant) have already been described (*Schaeffer et al., 2009*; *Schaeffer et al., 2012*). A gene encoding the elastase-resistant fragment of uromodulin (efUmod, lacking residues 27–294) was generated by overlapping PCRs (PCR1: residues 1–26; PCR2: residues 295–640) from wild-type uromodulin cDNA and inserted in pcDNA 3.1(+) (Life

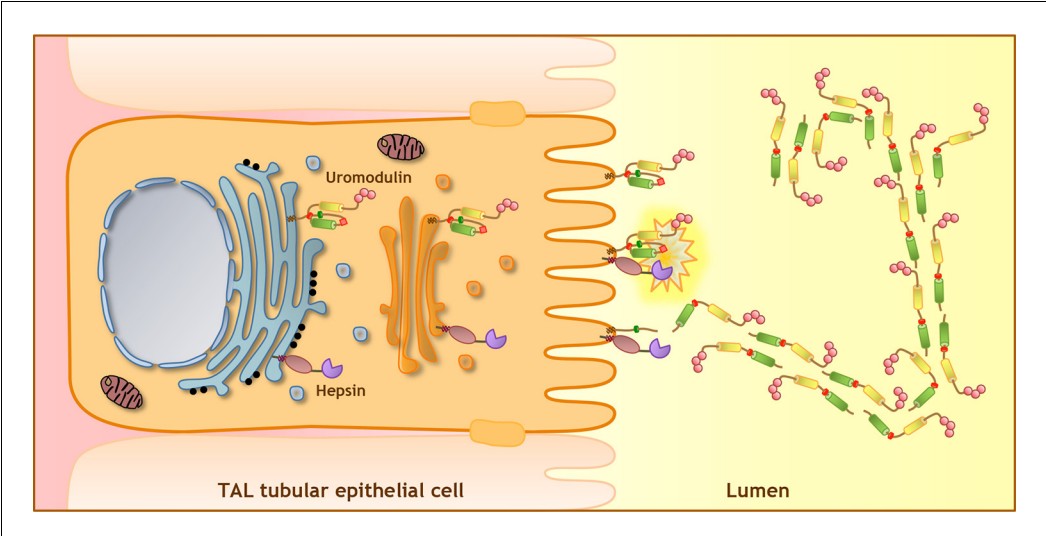

**Figure 8.** Model of uromodulin shedding and polymerisation. Uromodulin is exclusively expressed by TAL tubular epithelial cells. The protein enters the secretory pathway and reaches the plasma membrane in a polymerisation-incompetent conformation. This is ensured by the interaction between two hydrophobic patches within and next to the ZP-C subdomain (Internal Hydrophobic Patch, red circle, and External Hydrophobic Patch, dark green circle) (*Jovine et al., 2004*; *Schaeffer et al., 2009*; *Han et al., 2010*). Shedding by hepsin at the uromodulin consensus cleavage site (red diamond), likely occurring at the plasma membrane, releases the hydrophobic interaction, generating polymerisation-competent species that are assembled into polymeric filaments within the tubular lumen. Pink circles indicate N-terminal EGF-like domains, yellow and green cylinders represent ZP-N and ZP-C subdomains. The orientation of uromodulin within polymers is hypothetical.

Technologies, Carlsbad, CA) for expression studies in MDCK cells. To generate soluble efUmod for *in vitro* cleavage experiments, a DNA fragment encoding uromodulin residues 295–610 was cloned in pIRES-hrGFP II (Stratagene, Santa Clara, CA). Glycosylation site mutation N513Q and a C-terminal 6His-tag were inserted. The consensus cleavage site mutant $^{586}$RFRS$^{589}$ > $^{586}$AYAA$^{589}$ was also generated by site-directed mutagenesis (QuikChange Lightning Site-Directed mutagenesis kit, Agilent Technologies, Santa Clara, CA). Human hepsin and prostasin cDNA (OriGene, Rockville, MD) were cloned in pcDNA3.1/Zeo(+) (Life Technologies). Myc-tag was inserted at the C-terminus of hepsin by phosphorylating and annealing the following primers: forward 5'-CCGGTGAGCAAAAGCTGATTTC TGAGGAGGATCTGA-3' and reverse 5'- CCGGTCAGATCCTCCTCAGAAATCAGCTTTTGCTCA -3'. Catalytically inactive prostasin or hepsin were generated by mutating the serine residue of the catalytic triad (S238A in PRSS8, S353A in HPN) using the QuikChange Lightning Site-Directed mutagenesis kit (Agilent Technologies). Primer sequences for generation of all constructs are listed in *Table 1*.

Human HAI-1 cDNA (*Shimomura et al., 1997*) cloned in pcDNA3.1(+) was a kind gift of Prof. Clara Camaschella (San Raffaele Scientific Institute).

## Cell lines and culture conditions

Madin-Darby Canine Kidney (MDCK) cells and Human Embryonic Kidney 293 (HEK293) cells were grown in DMEM supplemented with 10% foetal bovine serum (Euroclone, Pero, Italy), 200 U/ml penicillin, 200 µg/ml streptomycin, and 2 mM glutamine at 37°C with 5% $CO_2$. They were transfected using Lipofectamine 2000 (Life Technologies) or Metafectene Pro (Biontex, Munich, Germany) respectively. Selection of stably transfected MDCK cells with G418 (Life Technologies) was started 24 hr after transfection and was pursued for 1–2 weeks. To obtain electrically tight monolayers (>100 Ω cm$^{-2}$), MDCK cells were grown on filters (Corning, Corning, NY) for 4 days. Uromodulin secretion in MDCK or HEK293 cells was analysed after 16 hr incubation in Optimem (Life Technologies).

## Cellular treatment with protease inhibitors

MDCK cells stably expressing uromodulin were grown on coverslip. After 24 h, the medium was replaced with Optimem supplemented with 0.1% protease inhibitor cocktail (PIC) (P8340, Sigma-Aldrich, Saint Louis, MO), 0.1% DMSO, 0.1 mM AEBSF, 0.08 µM aprotinin, 1.4 µM E64, 4 µM bestatin, 2 µM leupeptin and 1.5 µM pepstatin A (Sigma-Aldrich). Immunofluorescence analysis was performed after 24 hr of treatment.

## Silencing of hepsin and prostasin expression

Sfold (http://sfold.wadsworth.org/cgi-bin/index.pl) and siDESIGN center (http://www.thermoscientificbio.com/design-center/) were used to identify siRNAs targeting canine hepsin and prostasin. Selected oligos (sense sequence) are listed in *Table 2*. Oligonucleotides for shRNA strategy were designed combining the siRNA sequence (sense) followed by a loop (sequence 5'-3': TTCAAGAGA), the reverse complement of the siRNA sequence (antisense) and RNA polymerase III terminator (sequence 5'-3': TTTTTTGGAA). Oligos were phosphorylated, annealed and inserted in pENTR/ pTER+, a gift from Eric Campeau (plasmid # 17453, Addgene, Cambridge, MA) (*Campeau et al., 2009*). MDCK cells stably expressing uromodulin were co-transfected with pENTR/pTER+ vector and EGFP-expressing vector pcDNA3x(+)MyEGFP (Life Technologies). Cells were collected 24 hr after transfection and sorted using MoFlo sorter (Beckman Coulter, Brea, CA). Recovered EGFP-positive cells were maintained in culture for 48 hr.

## *In silico* analyses

A list of human serine proteases was derived from the MEROPS database (http://merops.sanger.ac.uk/). The expression profile of these enzymes in MDCK and HEK293 cells was analysed in Gene Expression Omnibus DataSets database (http://www.ncbi.nlm.nih.gov/gds) using baseline replicates of the following series: GSD3267 (*Hellman et al., 2008*), GSE20193 (*Sharma et al., 2010*), GSE18739 (*Abd Alla et al., 2010*), GSE15575 (*Abd Alla et al., 2009*), GSE1309 (*Zagranichnaya, 2005*), GSE1364 (*Jack et al., 2006*), GSE51118 and GDS1852 (*Elkon et al., 2005*). Membrane association of the enzymes was derived from Uniprot database (http://www.uniprot.org/). Protease

**Table 1.** Primers (5'-3') used to generate *UMOD* and protease constructs.

efUMOD-PCR1
T7: TAATACGACTCACTATAGGG
Reverse: ACACGTCCCCTCCACGTGGTGATGGTGATGATGAC

efUMOD-PCR2
Forward: CATCACCATCACCACGTGGAGGGGACGTGTGAGGA
BGHrev: TAGAAGGCACAGTCGAGG

efUMOD_sol
Forward: GGACAGATCTACGTCGACGGGACGTGTGAGGAGTGCAG
Reverse: CCGGAATTCCTAGTGATGATGGTGATGATGCTGGACACCTTTCCGTGTG

efUMOD_AYAA
Forward: CCTACCTGCTCTGGGACCGCATACGCAGCTGGGAGTGTCATAGATCAATCCC
Reverse: GGGATTGATCTATGACACTCCCAGCTGCGTATGCGGTCCCAGAGCAGGTAGG

efUMOD_N513Q
Forward: CACCCAGTAGCCAGGCCACGGACCCCC
Reverse: GGGGGTCCGTGGCCTGGCTACTGGGTG

PRSS8_S238A
Forward: GCCAGGGTGACGCTGGGGGCCCA
Reverse: TGGGCCCCCAGCGTCACCCTGGC

HPN_S353A
Forward: CTGCCAGGGCGACGCCGGTGGTCCCTTT
Reverse: AAAGGGACCACCGGCGTCGCCCTGGCAG

expression in TAL cells was analysed by including series GSE13672 (*Yu et al., 2009*) and GSE25223 (*Cheval et al., 2011*).

## Mouse lines

Wild-type, *Hpn*$^{-/-}$ (*Wu et al., 1998*), *Prss8*$^{lox/lox}$ (*Rubera et al., 2002*) and *Prss8*$^{-/-}$ animals are in C57BL/6J background. Adult nephron-specific *Prss8*-deficient (*Prss8*$^{-/-}$) mice were obtained by inter-breeding of *Prss8*$^{lox/lox}$ with *Pax8-rtTA/LC1*$^{tg/+}$ (*Traykova-Brauch et al., 2008*). Briefly, 28 day-old *Prss8*$^{-/-}$ and *Prss8*$^{lox/lox}$ (control) littermates were treated with doxycycline (2 mg/ml) in 2% sucrose in drinking water for 15 days. Genotyping by PCR was performed as previously described (*Malsure et al., 2014*; *Ronzaud et al., 2013*).

All animal procedures were carried out at the Lerner Research Institute, Cleveland Clinic, USA; at the University of Lausanne, Lausanne, Switzerland and at the University of Zurich, Zurich, Switzerland, according to protocols approved by the Institutional Animal Care and Use Committee at the Lerner Research Institute and by the Swiss Cantonal Veterinary Authority.

## Protein extracts, immunoprecipitation and Western blot analysis

MDCK and HEK293 cells, or mouse tissues were solubilised in lysis buffer (50 mM Tris-HCl, pH 7.4, 150 mM NaCl, 60 mM octyl-β-D-glucopyranoside and 0.1% PIC). Cell-conditioned medium or urinary proteins were precipitated with acetone and resuspended in PBS. When needed, samples were N-deglycosylated with PNGase F (New England Biolabs, Ipswich, MA). Lysates were quantified with the Bio-Rad Protein Assay (Bio-Rad, Hercules, CA).

Precleared lysates of HEK293 cells (300 µg) were loaded onto protein G Sepharose beads (Sigma-Aldrich) pre-conjugated with 2 µg of mouse anti-HA antibody (Covance, Princeton, NJ) and incubated 16 hr at 4°C. After washes in PBS-triton 0.5%, beads were resuspended in Laemmli Buffer.

Protein lysates, immunoprecipitated proteins, urinary or secreted proteins were separated on reducing 8% SDS-PAGE gel and transferred onto nitrocellulose membrane (GE Healthcare, Little Chalfont, United Kingdom). Western blotting (WB) was performed following standard protocols. Quantification was performed using the gel analysis option of ImageJ software (http://rsbweb.nih.gov/ij/) (*Schneider et al., 2012*).

**Table 2.** Oligonucleotide sequences used to generate shRNAs.

| Target gene | Target position from ATG codon | siRNA oligo (5'-3') |
| --- | --- | --- |
| HPN_siRNA#1 | 122-140 | CCTTCCTACTCAAGAGTGA |
| HPN_siRNA#2 | 1195-1213 | TGGATCTTCCAGGCCATAA |
| PRSS8_siRNA#1 | 258-276 | GAAGGAAGACTATGAGGTA |
| PRSS8_siRNA#2 | 593-611 | TGTACAACATCAACGCTAA |
| Control siRNA | NA | TAGTGAGATTCGTTAAGAT |

## Collection and processing of plasma, urine samples and renal tissues

Kidneys for immunofluorescence or microdissection of nephron segments were collected from 2–3-month-old male wild-type control mice. Nephron segment microdissection was performed and validated as previously described (*Glaudemans et al., 2014*). Plasma, urine and kidneys from wild-type and $Hpn^{-/-}$ mice were collected from 2.5-month-old male mice. Urine and kidneys from $Prss8^{lox/lox}$ and $Prss8^{-/-}$ were collected from 1.5-month-old mice.

Animals were housed in a light- and temperature-controlled room with *ad libitum* access to tap water and standard chow. Urine collections (16 h) were obtained at baseline using individual metabolic cages, after appropriate training. Urinary creatinine quantification was performed using creatinine assay kit (Cayman Chemical, Ann Arbor, MI). Blood samples were collected from the vena cava. Serum creatinine levels were measured by Consolidated Veterinary Diagnostics (West Sacramento, CA). Polymerisation assay on urinary samples was performed as previously described (*Jovine et al., 2002*). Supernatants were precipitated with acetone, while pellets were solubilised in Laemmli Buffer.

## Immunofluorescence analysis

Immunofluorescence (IF) of cells grown on coverslip was carried out as previously described (*Schaeffer et al., 2009*). Cells grown on filters were fixed 20 min in 4% paraformaldehyde (VWR International, Radnor, PA) and permeabilised 15 min at 37°C with 0.7% gelatin and 0.016% saponin. Cells were incubated for 1 hr at 37°C with primary antibody and subsequently with the appropriate AlexaFluor-labeled secondary antibody (Life Technologies). Nuclei were stained with 4',6-diamidino-2-phenylindole (DAPI, Life Technologies) and slides were mounted using fluorescent mounting medium (Dako, Agilent Technologies). Slides were visualised under a DM 5000B fluorescence upright microscope (Leica, Wetzlar, Germany) or under an UltraVIEW ERS spinning disk confocal microscope (PerkinElmer, Waltham, MA). Quantification of mean polymeric area was performed using ImageJ software, as previously described (*Schaeffer et al., 2012*). Polymers were quantified on 5–9 independent images for each condition taken at 10x magnification. Cell and polymer reconstructions were carried out by collecting stacks of consecutive confocal images at 0.5 µm intervals. Extended focus 2D images were obtained by using Volocity 3D Image Analysis Software version 6.3 (PerkinElmer, Waltham, MA).

Kidneys for immunofluorescence were collected from mice anaesthetised and perfused with 3% paraformaldehyde and snap-frozen in liquid propane. Kidney sections (4–5 µm thick) were reactivated in PBS for 30 min and blocked with 10% goat serum for 30 min. They were incubated overnight at 4°C with with relevant primary antibodies (anti-hepsin, -prostasin or –uromodulin). For co-localisation with uromodulin, sections were blocked again for 30 min and incubated for 2 hr with the anti-uromodulin primary antibody. Slides were incubated with the appropriate AlexaFluor-labelled secondary antibody diluted in PBS containing DAPI and mounted with Prolong Gold anti-fade reagent (Invitrogen, Life Technologies). Slides were visualised with a Leica SP5 Confocal microscope.

## Antibodies

We used the following primary antibodies: mouse anti-HA (MMS-101P, 1:1000 for WB and 1:500 for IF, Covance), rabbit anti-FLAG (F7425, 1:500 for IF, Sigma-Aldrich), goat anti-Myc (NB600-335, 1:500 for IF, Novus Biologicals, Littleton, CO), sheep anti-uromodulin (T0850, 1:1000 for WB, US Biological, Salem, MA), sheep anti-uromodulin (K90071C, 1:200 for IF, Meridian Life Science, Cincinnati,

OH), goat anti-uromodulin (55140, 1:1000 for WB and 1:500 for IF, MP Biomedicals, Santa Ana, CA), rabbit anti-hepsin (100022, 1:1000 for WB and 1:50 for IF, Cayman Chemical), sheep anti-prostasin (AF4599, 1:1000 for WB and 1:200 for IF, R&D System, Minneapolis, MN), rabbit anti-prostasin (kind gift of Prof. Carl Chai, University of Central Florida College of Medicine, FL; 1:200 for IF) (Chen, 2006), rabbit anti-HAI-1 (H-180, 1:1000 for WB, Santa Cruz Biotechnology, Santa Cruz, CA), rabbit anti-PDI (H-160, 1:1000 for WB, Santa Cruz Biotechnology), mouse anti-E-cadherin (610182, 1:500 for IF, BD Biosciences, San Jose, CA), mouse anti-KDEL (ADI-SPA-827-D, 1:500, Enzo Life Sciences, Farmingdale, NY), mouse anti-GAPDH (6C5, 1:5000 for WB, Santa Cruz Biotechnology), mouse anti-β actin (A2228, 1:16000 for WB, Sigma-Aldrich), mouse anti-α tubulin (SC-8035, 1:1000 for WB, Santa Cruz Biotechnology) and mouse anti-5His (34660, 1:1000 for WB, Qiagen, Venlo, The Netherlands).

## RNA extraction, RT-PCR and Real-Time qPCR

Total RNA was extracted from confluent cells or from mouse kidneys using TRIzol reagent (Life Technologies) and reverse-transcribed using iScript kit (Bio-Rad). Expression of target genes was analysed by RT-PCR (@Taq, Euroclone) or by Real-Time qPCR on LightCycler 480 (Roche, Basel, Switzerland) using qPCR Core kit for SYBR Assay (Eurogentec, Seraing, Belgium).

Hpn and Prss8 expression in microdissected nephron segments were assessed by Real-Time qPCR with a CFX96™ Real-Time PCR Detection System (Bio-Rad) using iQ™ SYBR Green Supermix (Bio-Rad). The relative mRNA expression of genes of interest was calculated following the ΔΔCt method (Pfaffl, 2001). Primer sequences are listed in Table 3 and Table 4.

## Statistical analysis

Comparisons between groups were performed using two-tailed unpaired t-test or Mann-Whitney test. We expressed continuous measures as the mean ± the standard deviation (s.d.); averages of measurements as the mean ± the standard error of the mean (s.e.m.). We set significance level to $p < 0.05$.

## Purification of efUmod and in vitro cleavage

Chinese Hamster Ovary (CHO) DG44 cells were grown in α-MEM medium supplemented with 10% foetal bovine serum (Life Technologies) and stably transfected with pIRES-hrGFP_efUMOD. Stable cells were cultured in HyQ SFM4CHO-utility medium (Thermo Scientific, Waltham, MA). efUmod secreted by CHO cells was purified by batch IMAC with Ni-NTA Superflow (Qiagen), followed by SEC using a HiLoad 26/60 Superdex 200/HiPrep 26/60 Sephacryl 200 double column system (GE Healthcare) equilibrated against 10 mM Tris-HCl pH 8.0 at 4°C, 50 mM NaCl, and concentration. Recombinant efUmod (25 ng/μl) was incubated with human serine proteases hepsin (Enzo Life Sciences, Farmingdale, NY) or prostasin (R&D System) in 50 mM Tris, 20 mM NaCl, pH 7.8 at 37°C for 10 hr. Picomolar ratio between protease and efUmod was 1:100 for hepsin and 1:5 for prostasin.

## Electron microscopy of uromodulin filaments secreted by MDCK cells

MDCK cell conditioned media were centrifuged 30 min at 18,000 x g at 4°C, resuspended in ultrapure water and applied to glow-discharged 400 mesh copper grids with a carbon support film. After 5 min the grids were washed with ultrapure water for 2 min and then negatively stained with 2%

**Table 3.** Primers used for gene expression analysis (RT-PCR).

| Gene | PCR product length | Primer (5'-3') |
| --- | --- | --- |
| HPN (Dog / Human) | 353 bp | Forward: GCTGCGAGGAGATGGGCTTC<br>Reverse: CGGGAAGCAGTGGGCGGCTG |
| PRSS8 (Dog / Human) | 547 bp | Forward: CCTGGCAGGTCAGCATCACC<br>Reverse: CCAGAGTCACCCTGGCAGGC |
| GAPDH (Human) | 314 bp | Forward: CCACCCAGAAGACTGTGGAT<br>Reverse: GTTGAAGTCAGAGGAGACCACC |
| GAPDH (Dog) | 289 bp | Forward: CTCTGGGAAGATGTGGCGTGAC<br>Reverse: GTTGAAGTCACAGGAGACCACC |

**Table 4.** Primers used for gene expression analysis (Real-Time qPCR).

| Gene | PCR product length | Primer (5'-3') |
|---|---|---|
| HPN (Dog) | 164 bp | Forward: TGGTCCACCTGTCCAGCCCC<br>Reverse: GACTCGGGCCTCCTGGAGCA |
| Hpn (Mouse) | 152 bp | Forward: CTGACTGCTGCACATTGCTT<br>Reverse: GGGTCTCGAAAGGGAAGGTA |
| PRSS8 (Dog) | 156 bp | Forward: TCCGGACTTGCCTTCTGCGGT<br>Reverse: AGCTGAGAGCACCCACTGCTCA |
| GAPDH (Dog) | 289 bp | Forward: CTCTGGGAAGATGTGGCGTGAC<br>Reverse: GTTGAAGTCACAGGAGACCACC |
| Prss8 (Mouse) | 147bp | Forward: ATCACCCACTCAAGCTACCG<br>Reverse: AGTACAGTGAAGGCCGTTGG |
| Umod (Mouse) | 207 bp | Forward: ATGGACCAGTCCTGTCCTG<br>Reverse: CCGTCTCGCTTCTGTTGAGT |
| Hprt1 (Mouse) | 162 bp | Forward: ACATTGTGGCCCTCTGTGTG<br>Reverse: TTATGTCCCCCGTTGACTGA |

uranyl acetate. Samples were analysed using a CM120 electron microscope (Philips, Eindhoven, The Netherlands) equipped with a LaB$_6$ electron source. Images were recorded on Kodak SO163 electron film (Eastman Kodak, Rochester, NY) and digitised using an Epson Perfection 4990 PHOTO flatbed scanner (Epson, Suwa, Japan).

## Mass spectrometric analysis of uromodulin secreted by MDCK cells

MDCK cell-conditioned media were centrifuged at 3,000 x $g$ for 5 min at 4°C. Supernatants were concentrated using Amicon Ultra-3K (Millipore, Billerica, MA) at 4,000 x $g$ at 4°C and depleted of the most abundant proteins using IgY-12 columns (Genway, San Diego, CA). Proteins were precipitated with 60% TCA (trichloroacetic acid, Sigma-Aldrich) for 1 hr at -20°C and washed with 90% acetone. Two hundred μg were diluted in 130 μl of a buffer containing 5 M urea (Sigma-Aldrich), 2 M thiourea (Sigma-Aldrich), 2% CHAPS (Sigma-Aldrich), 2% Zwittergent (GE Healthcare), 100 mM DeStreak (GE Healthcare) and 0.5% IPG buffer pH 3–10 NL (GE Healthcare) and loaded on Immobiline Dry strip pH 3–10 NL, 7 cm (GE Healthcare, total focusing run 50,000 Vh). IPG strips were sequentially reduced and alkylated prior to the second dimension electrophoresis. SDS-PAGE 8% gels were Coomassie stained and spots of interest were excised and in-gel digested as previously reported (*Shevchenko et al., 1996*). For nLC-MS/MS analysis, peptide mixtures were acidified up to 1% formic acid and analysed on an API QStar PULSAR (AB-Sciex, Framingham, MA) mass spectrometer as previously reported (*Magagnotti et al., 2013*). All MS/MS samples were analysed using MASCOT engine (version 2.2.07, Matrix Science, London, United Kingdom) and X!Tandem (within Scaffold software, v.3.6.4, Proteome Software, Portland, OR) to search the UniProt_CP_Human 2013_05 database. Peptide mass tolerance of 200 ppm and 0.6 Da for precursor and fragment ions were selected respectively. Searches were performed with: semi-Lys-C or semi-Asp-N specificity; carbamidomethylation of cysteine as fixed modification and oxidation of methionine as variable modification. Scaffold was used to validate MS/MS based peptide and protein identifications with protein thresholds set to 99%, 2 peptides minimum and peptide thresholds set to 95% minimum.

## Mass spectrometric analysis of urinary uromodulin

Uromodulin was purified from 1 ml of mouse urinary samples as previously described (*Santambrogio et al., 2008*). PNGase F-treated and alkylated samples were separated on 8% SDS–PAGE gel. Bands of interests were excised from Coomassie stained gels, reduced, alkylated and finally digested overnight with trypsin or Asp-N (Roche) (*Santambrogio et al., 2008*). After acidification, peptide mixtures were concentrated and desalted on homemade Stagetips μC18 (*Rappsilber et al., 2007*) and injected in a capillary chromatographic system (EasyLC, Proxeon Biosystems, Thermo Scientific). Peptide separations occurred on a homemade fused silica capillary column (75 μm i.d. × 25 cm), packed with 3 μm ReproSil-Pur C18-AQ (Dr. Maisch, Ammerbuch-Entringen, Germany). A gradient of eluents A (H$_2$O with 2% acetonitrile, 0.5% acetic acid) and B

(80% acetonitrile with 0.5% acetic acid) was used to achieve separation, from 4% to 70% B (in 65 min, 0.15 µL/min flow rate). The LC system was connected to an LTQ-Orbitrap mass spectrometer (Thermo Scientific) equipped with a nanoelectrospray ion source (Proxeon Biosystems). MS and MS/MS spectra were acquired selecting the ten most intense ions per survey spectrum acquired in the orbitrap from $m/z$ 300–1750 with 60,000 resolution. Target ions selected for the MS/MS were fragmented in the ion trap and dynamically excluded for 120 sec. Target values were 1,000,000 for survey scan and 100,000 for MS/MS scan. MS/MS spectra were converted into peaklist (.msm files) and analysed using MASCOT and Scaffold searching against the UniProt_CP_Mus 2012_10 database. Searches were performed with: semi-Asp-N or semi-trypsin specificity; parent ion tolerance of 5.0 ppm and fragment ion mass tolerance of 0.60 Da; N-ethylmaleimide or carbamidomethyl of cysteine as fixed modifications; oxidation of methionine, asparagine deamidation and acetylation of the N-terminus of proteins as variable modifications.

## Acknowledgements

We thank H Debaix, N Goelz, Z Guo, N Sidenius, C Camaschella, L Silvestri, D Talarico, R Vago, N Hooper, L Chen, KX Chai, M Bochud, C Hayward, N Mancini, N Clementi, M Trudu, A Cretore, the San Raffaele Alembic and Fractal facilities for help, technical assistance and fruitful discussions.

## Additional information

### Funding

| Funder | Grant reference number | Author |
| --- | --- | --- |
| Fondazione Telethon | TCR08006 | Luca Rampoldi |
| Fondazione Telethon | GGP14263 | Luca Rampoldi |
| Ministero della Salute | RF-2010-2319394 | Luca Rampoldi |
| Swiss National Science Foundation | 31003A-144198/1 | Edith Hummler |
| National Center of Competence in Research | Kidney.Ch | Edith Hummler Olivier Devuyst |
| Fonds National de la Recherche Luxembourg | 6903109 | Eric Olinger |
| European Commission Seventh Framework Programme (FP7/2007-2013) | 246539 (Marie Curie Actions Programme), 305608 (EURenOmics) | Olivier Devuyst |
| Fonds De La Recherche Scientifique - FNRS | | Olivier Devuyst |
| Fonds De La Recherche Scientifique Medicale | | Olivier Devuyst |
| Gebert Rüf Stiftung | GRS-038/12 | Olivier Devuyst |
| Swiss National Science Foundation | 310030-146490 | Olivier Devuyst |
| Swiss National Science Foundation | 32003B-149309 | Olivier Devuyst |
| Swedish Research Council | 2012-5093 | Luca Jovine |
| Goran Gustafsson Foundation for Research in Natural Sciences and Medicine | | Luca Jovine |
| European Research Council, under the European Union's Seventh Framework Programme (FP7/2007-2013) | ERC 260759 | Luca Jovine |

The funders had no role in study design, data collection and interpretation, or the decision to submit the work for publication.

## Author contributions

MBr, Conception and design, Acquisition of data, Analysis and interpretation of data, Drafting or revising the article; SP, LH, AC, FC, AA, CS, EO, JP, SS, RP, SL, MBo, Acquisition of data, Analysis and interpretation of data; AB, EH, OD, QW, LJ, LR, Conception and design, Analysis and interpretation of data, Drafting or revising the article

## Ethics

Animal experimentation: All animal studies were performed in strict adherence with the NIH Guide for the Care and Use of Laboratory Animals. Experimental procedures and animal maintenance at the University of Lausanne followed federal guidelines and were approved by local authorities (Service de la consommation et des affaires vétérinaires, authorization numbers 1003.7 and 25520 for animal experimentation, and VD-H06 for animal housing). Animal studies at the University of Zurich were performed under the approval of the Swiss Cantonal Veterinary Authority (Number: 103/2014). The protocol was approved by the Institutional Animal Care and Use Committee (IACUC) of the Cleveland Clinic (Number: 2015-1403).

# Additional files

## Major datasets

The following previously published datasets were used:

| Author(s) | Year | Dataset title | Dataset URL | Database, license, and accessibility information |
|---|---|---|---|---|
| Hellman NE, Spector J, Robinson J, Zuo X, Saunier S, Antignac C, Tobias JW, Lipschutz JH | 2008 | Madin-Darby canine kidney type II cell line response to hepatocyte growth factor | http://www.ncbi.nlm.nih.gov/sites/GDSbrowser?acc=GDS3267 | Publicly available at NCBI Gene Expression Omnibus (GDS3267) |
| Sharma GG, So S, Gupta A, Kumar R, Cayrou C, Avvakumov N, Bhadra U, Pandita RK, Porteus MH, Chen DJ, Cote J, Pandita TK | 2010 | Altered levels of MOF and decreased levels of H4K16ac correlate with a defective DNA damage response (DDR) | http://www.ncbi.nlm.nih.gov/geo/query/acc.cgi?acc=GSE20193 | Publicly available at NCBI Gene Expression Omnibus (GSE20193) |
| Abd Alla J, Pohl A, Reeck K, Streichert T, Quitterer U | 2009 | Establishment of a novel model to assess the function of proteins under in vivo conditions | http://www.ncbi.nlm.nih.gov/geo/query/acc.cgi?acc=GSE18739 | Publicly available at NCBI Gene Expression Omnibus (GSE18739) |
| Abd Alla J, Reeck K, Streichert T, AbdAlla S, Quitterer U | 2009 | Expression data from human embryonic kidney cells (HEK293) cultivated in high and low glucose | http://www.ncbi.nlm.nih.gov/geo/query/acc.cgi?acc=GSE15575 | Publicly available at NCBI Gene Expression Omnibus (GSE15575) |
| Zagranichnaya TK, Wu X, Danos A, Villereal ML | 2004 | Gene expession profiles for monoclonal cell lines with high or low levels of store-operated Ca2+ entry. | http://www.ncbi.nlm.nih.gov/geo/query/acc.cgi?acc=GSE1309 | Publicly available at NCBI Gene Expression Omnibus (GSE1309) |
| Mead EA, Garst J, DeSantis AM, Slaughter SM, Jervis J, Jack G, Yoon JH, Helm RF, Potts MA | 2005 | Long-Term Storage of Human Cells at Ambient Temperature | http://www.ncbi.nlm.nih.gov/geo/query/acc.cgi?acc=gse1364 | Publicly available at NCBI Gene Expression Omnibus (GSE1364) |
| Byun JS | 2013 | p300 is a preferred client protein for Nuclear Chaperones | http://www.ncbi.nlm.nih.gov/geo/query/acc.cgi?acc=GSE51118 | Publicly available at NCBI Gene Expression Omnibus (GSE51118) |

| | | | | |
|---|---|---|---|---|
| Elkon R, Rashi-Elkeles S, Lerenthal Y, Linhart C, Tenne T, Amariglio N, Rechavi G, Shamir R, Shiloh Y | 2005 | siRNA knockdown and transcriptional response to DNA damage | http://www.ncbi.nlm.nih.gov/sites/GDSbrowser?acc=GDS1852 | Publicly available at NCBI Gene Expression Omnibus (GDS1852) |
| Yu MJ, Miller RL, Uawithya P, Rinschen MM, Khositseth S, Braucht DW, Chou CL, Pisitkun T, Nelson RD, Knepper MA | 2009 | Mouse mpkCCD cells, Rat Kidney Proximal Tubule, and Rat Kidney Medullary Thick Ascending Limb | http://www.ncbi.nlm.nih.gov/geo/query/acc.cgi?acc=GSE13672 | Publicly available at NCBI Gene Expression Omnibus (GSE13672) |
| Cheval L, Pierrat F, Dossat C, Genete M, Imbert-Teboul M, Duong van Huyen J, Poulain J, Wincker P, Weissenbach J, Piquemal D | 2010 | Atlas of gene expression in the mouse kidney | http://www.ncbi.nlm.nih.gov/geo/query/acc.cgi?acc=GSE25223 | Publicly available at NCBI Gene Expression Omnibus (GSE25223) |

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
