## [Decision Letter]

Thank you for submitting your work entitled "The serine protease hepsin mediates urinary secretion and polymerisation of Zona Pellucida domain protein uromodulin" for peer review at *eLife*. Your submission has been favorably evaluated by Tony Hunter (Senior editor and Reviewing editor) and two reviewers.

The Reviewing editor has drafted this decision to help you prepare a revised submission.

We have received reviews from two experts, who found your observations to be a significant advance in our understanding of how uromodulin, in particular, and potentially other ZP proteins are processed. However, Reviewer #2 and the BRE (Board of Reviewing Editors) member both felt that the paper would be significantly strengthened by the addition of evidence that this function of hepsin is physiologically important. For instance, if your major conclusion is correct, then one would expect to observe some renal effects in the hepsin knockout mice. For instance, do these mice exhibit increased urinary tract infection or other predictable renal phenotypes? If you can include data along these lines and address the other points raised by Reviewer 1, then we would consider a revised version of your paper for publication in *eLife*.

*Reviewer #1:*

This is an interesting and exciting paper that provides very convincing evidence that hepsin (TMPRSS1) is the protease that regulates the proteolytic release of the ZP domain protein uromodulin from the apical plasma membrane of renal tubule cells, thereby allowing its polymerization. The work has been performed to a high standard, and the findings represent a significant advance that should be of wide interest considering the large number of ZP-domain proteins with diverse functions that have been identified. I have, however, a few questions and comments the authors could address.

1) What is known about the longer and more abundant isoform of uromodulin that is secreted by transfected MDCK cells? How has it been secreted if it has not been released from the membrane? Is the sample shown in Figure 1 from the medium?

2) PIC and some of the component proteases certainly reduce the numbers of cells that are expressing uromodulin polymers on their surface. Do the inhibitors affect overall protein expression levels in any way? Could it be that many of the cells are compromised in some fashion?

3) Figure 3: It is hard to see that uromodulin polymerization only occurs when wild type proteases are expressed. The images are not that convincing and could be improved. Some confocal images, and possibly some z-projections, may help here.

4) In the first paragraph of the subsection “Hepsin mediates physiological cleavage and polymerisation of uromodulin in vivo”: What is meant by the 'mature' form of uromodulin? And what is the longer isoform shown in Figure 7 (right)? Is this partially N-deglycosylated? The text reads as if the short uromodulin isoform that is seen in the *Hpn^-/-^* mice is the same as that seen in the wild-type mice, when it is presumably not, as it lacks the physiological C-terminus. Some clarification is required here.

5) Is prostasin expressed on the surface of the TAL in mouse kidney and does it co-localize with uromodulin?

6) It is certainly intriguing that the phenotype of the tectorial membrane in the inner ear of *Hpn^-/-^* mice is morphologically very similar (at the light microscope level) to that seen in mice that are heterozygous for deafness-causing, semi-dominant missense mutations in the ZP domain of Tecta. Mutations in ZP2 corresponding to two of these Tecta ZP domain mutations (C1837G and Y1870C) have been shown to prevent the secretion of ZP2 in vitro (Jovine et al., 2002).

*Reviewer #2:*

The biochemistry and discovery of the protease responsible for cleaving uromodulin are compelling. However, it is disappointing that there wasn't any evidence to suggest hepsin is physiologically important. Based on their data showing reduced secretion of uromodulin – one would expect to see some renal effect in the hepsin KO mice. Such evidence would make the paper much more exciting.

---

## [Author Response]

*We have received reviews from two experts, who found your observations to be a significant advance in our understanding of how uromodulin, in particular, and potentially other ZP proteins are processed. However, Reviewer #2 and the BRE (Board of Reviewing Editors) member both felt that the paper would be significantly strengthened by the addition of evidence that this function of hepsin is physiologically important. For instance, if your major conclusion is correct, then one would expect to observe some renal effects in the hepsin knockout mice. For instance, do these mice exhibit increased urinary tract infection or other predictable renal phenotypes? If you can include data along these lines and address the other points raised by Reviewer 1, then we would consider a revised version of your paper for publication in* eLife.

Reviewer #1:

*1) What is known about the longer and more abundant isoform of uromodulin that is secreted by transfected MDCK cells? How has it been secreted if it has not been released from the membrane? Is the sample shown in Figure 1 from the medium?*

We apologise if this was not clear. The sample in Figure 1 is indeed from the medium of MDCK cells stably expressing wild-type human uromodulin. Uromodulin is secreted as two isoforms that can be clearly separated by gel electrophoresis after deglycosylation. The short isoform has the same C-terminus and polymerisation properties of urinary protein. Since our main interest was to identify the protease releasing urinary uromodulin, we focused our work on the mechanism generating such short isoform. This was indeed a key step for the final identification of hepsin. We have modified the text in the Results section to clarify this point (subsection “Uromodulin cleavage and polymerisation in MDCK cells is mediated by a serine protease”).

We have not investigated the nature of the process generating the long isoform, since we considered this beyond the scope of our study and its physiological relevance is uncertain. The long isoform is likely released from the membrane by a different proteolytic or phospholipase-mediated cleavage.

We carried out mass spectrometry analysis on this isoform revealing a C-terminus (mapped after Asp-N digestion at A611) that is more distal then the physiological one (F587, short isoform) (these data, not included in the manuscript, are shown in Figure 9). This is consistent with the slower electrophoretic mobility of this isoform (Figure 1).

Author response image 1.Mass spectrometry analysis of the long uromodulin isoform released by MDCK cells.(**A**) Mass spectrometry sequence coverage (42%) of Asp-N-digested human uromodulin (long isoform) from the medium of stably expressing MDCK cells. The analysis was carried out as described in the Methods section of the manuscript. The long isoform was separated in 2D electrophoresis. Matching peptides are shown in red, while the C- terminal peptide is shown in blue. This peptide ends at A611, a distal C-terminal relative to the one identified for the short isoform (F587) that is identical to the one reported for human urinary uromodulin (Santambrogio et al., 2008). (**B**) Representative tandem mass-spectrometry (MS/MS) spectrum confirming the identity of the ^594^DQSRVLNLGPITRKGVQA^611^ C-terminal peptide of the long uromodulin isoform released by MDCK cells and table of fragmented ions.**DOI:**
http://dx.doi.org/10.7554/eLife.08887.030

The presence of a more distal cleavage in MDCK cells is reminiscent of what we identified in the urine of *Hpn* knock-out mice. In vivouromodulin misprocessing at the plasma membrane is only observed in the absence of the physiological protease, i.e. hepsin. In MDCK cells, this prevalent component likely reflects weak expression of hepsin in these cells.

*2) PIC and some of the component proteases certainly reduce the numbers of cells that are expressing uromodulin polymers on their surface. Do the inhibitors affect overall protein expression levels in any way? Could it be that many of the cells are compromised in some fashion?*

We thank the reviewer for this comment. We added this control showing that PIC treatment, that includes the single inhibitors at the same concentration at which they were tested alone, does not affect protein expression and intracellular localisation, as shown by Western blot and immunofluorescence analysis (see new Figure 2—figure supplement 1). We conclude that reduction of polymers on the membrane of MDCK cells is not due a reduced amount of protein reaching the plasma membrane, but rather by inhibition of its physiological cleavage (subsection “Uromodulin cleavage and polymerisation in MDCK cells is mediated by a serine protease”, third paragraph).

*3) Figure 3: It is hard to see that uromodulin polymerization only occurs when wild type proteases are expressed. The images are not that convincing and could be improved. Some confocal images, and possibly some z-projections, may help here.*

We agree with the reviewer that images showing uromodulin polymerisation in HEK293 cells expressing wild-type proteases could be unclear (the reviewer is referring to Figure 3). This was partly due to the fact that the original resolution of the pictures was reduced after conversion into pdf format. Nevertheless, as suggested by the reviewer, we have repeated the experiment and acquired high-resolution confocal images of consecutive stacks to generate extended focus images of cell surface and uromodulin polymers.

This analysis contributed to significantly improve the quality of images that are now more clearly showing the presence of polymers only when wild type proteases are expressed (see new Figure 3).

*4) In the first paragraph of the subsection “Hepsin mediates physiological cleavage and polymerisation of uromodulin in vivo“Line 175: What is meant by the 'mature' form of uromodulin? And what is the longer isoform shown in Figure 7 (right)? Is this partially N-deglycosylated? The text reads as if the short uromodulin isoform that is seen in the* Hpn^-/-^*mice is the same as that seen in the wild-type mice, when it is presumably not, as it lacks the physiological C-terminus. Some clarification is required here.*

We thank the reviewer for this comment and agree that this part needed some clarification.

In mice lacking hepsin, uromodulin is misprocessed and its urinary secretion is reduced. Such reduction is not due to changes in gene expression (Figure 6) nor to defective protein delivery to the apical membrane. The latter is demonstrated by the fact that the protein accumulating in the kidneys of *Hpn^-/-^* is the “mature” isoform, i.e. it carries post-Golgi type of glycosylation as opposed to the lower band corresponding to the ER precursor (the Endo H sensitivity of the two uromodulin isoforms that are detected in Western blot on mouse kidney lysates was previously shown in Bernascone et al., Hum. Mol. Genet.2010). Importantly, we now added immunofluorescence analysis of uromodulin distribution in mouse kidneys showing that uromodulin signal is clearly enriched on the luminal membrane of TAL segments (new Figure 6).

In the absence of hepsin, uromodulin is clearly misprocessed and it is released in the urine as two isoforms that can be separated on gel electrophoresis after deglycosylation. The shorter one has electrophoretic migration similar to the isoform in wild-type urine, though it bears a different, more distal C-terminus (Figure 7). We have now added mass-spectrometry analysis of the longer isoform, confirming that it is not a product of partial deglycosylation, but rather a product of protein misprocessing with an even more distal C-terminus (new Figure 1).

The part of the Results section describing these findings (Figure 6 and Figure 7) has been rephrased and integrated to improve its clarity (subsection “Hepsin mediates physiological cleavage and polymerisation of uromodulin in vivo”, first two paragraphs).

*5) Is prostasin expressed on the surface of the TAL in mouse kidney and does it co-localize with uromodulin?*

As suggested by the reviewer, we checked expression of prostasin in the TAL segment of mouse kidney by Real-time qRT-PCR on RNA from microdissected nephron segments and immunofluorescence. Prostasin is weakly expressed in the TAL (while for instance it is highly expressed in proximal tubules) where it is mostly detected on the apical membrane of epithelial cells, as shown by co-localisation with uromodulin (see new Figure 7—figure supplement 2 and corresponding description in the subsection “Hepsin mediates physiological cleavage and polymerisation of uromodulin in vivo”, last paragraph). These results are consistent with expression data derived from available transcriptomes (Figure 3) and with localisation of transfected prostasin on the apical membrane of polarised MDCK cells (Figure 5).

Despite co-localisation of prostasin and uromodulin in TAL cells, this protease is not involved in physiological processing of uromodulin, as clearly demonstrated by the analysis of uromodulin processing in the specific knock-out mice (Figure 7—figure supplement 2).

*6) It is certainly intriguing that the phenotype of the tectorial membrane in the inner ear of* Hpn^-/-^*mice is morphologically very similar (at the light microscope level) to that seen in mice that are heterozygous for deafness-causing, semi-dominant missense mutations in the ZP domain of Tecta. Mutations in ZP2 corresponding to two of these Tecta ZP domain mutations (C1837G and Y1870C) have been shown to prevent the secretion of ZP2 in vitro (Jovine et al., 2002).*

We thank the reviewer for this interesting comment. The morphological similarity of structural abnormalities of tectorial membrane in *Hpn* knock-out mice and in mouse models carrying deafness-causing *Tecta* mutations (Legan et al., 2005, 2014) indeed suggests the intriguing possibility that hepsin could play a role in tectorin processing. One could hypothesise that structural defects of the tectorial membrane in these models are due to reduced/altered secretion of tectorins, either because of protein trafficking defect (*Tecta* semi-dominant mutations) or because of misprocessing at the plasma membrane (*Hpn* loss-of-function mutation). We rephrased this part of the Discussion section and added citation of these papers (last paragraph).

Reviewer #2:

*The biochemistry and discovery of the protease responsible for cleaving uromodulin are compelling. However, it is disappointing that there wasn't any evidence to suggest hepsin is physiologically important. Based on their data showing reduced secretion of uromodulin – one would expect to see some renal effect in the hepsin KO mice. Such evidence would make the paper much more exciting.*

We thank the reviewer for considering the experimental data presented in the manuscript a compelling demonstration that hepsin is the protease responsible for uromodulin physiological cleavage. Considering that this work spans extensive *in vitro* and *in vivo* analyses, we believe that this conclusion is *per se* physiologically relevant, as it identifies the first *in vivo* substrate of hepsin and the first protease involved in the extracellular release of a ZP domain protein.

We understand the point raised by the reviewer (also shared by the BRE). If the lack of physiological processing of uromodulin is associated to a renal phenotype is indeed an interesting question, and we have been extensively thinking of ways to address it.

However, we feel that experiments along this line in *Hpn*^-/-^ mice would likely be inconclusive for the following reasons:

1) Currently uromodulin biological functions are not clearly determined. The protective role of urinary uromodulin against UTI and nephrocalcinosis has been established by studies on *Umod* KO mice, challenged by transuretheral bacteria injection (Bates et al., Kidney Int 2004; Mo et al., Am J Physiol Ren Physiol2004) or by ethylene glycol diet (Mo et al., Kidney Int2004). Spontaneous nephrocalcinosis was also reported, with frequency increasing with age (Mo et al., Am J Physiol Ren Physiol 2007; Liu et al., Am J Physiol Ren Physiol 2010). Although with some contrasting results, possibly due to different genetic background, baseline renal function parameters in *Umod* mice are not substantially affected, with the possible exception of reduced creatinine clearance, to date unexplained (Bachmann et al., Am J Physiol Ren Physiol 2005; Graham et al., Hypertension 2013).

Predictable renal phenotypes in *Hpn* KO mice would hence be based on uromodulin functions as deduced from *Umod* KO studies. However, *Hpn* KO mice show misprocessing and reduction of 60-70% of urinary uromodulin levels that may not mimic complete absence of urinary protein and hence lead to the same phenotype. It is of note that phenotype in *Umod*^+/-^ mice (possibly closer to *Hpn* KO mice in terms of levels of uromodulin urinary) has not been reported, with the exception of one study showing increased frequency of nephrocalcinosis in aged mice (Liu et al., Am J Physiol Ren Physiol 2010).

2) The interpretation of any experiment assessing renal phenotype(s) in *Hpn* KO mice would be undermined by the fact that any evidence could not be conclusively linked to reduced/misprocessed uromodulin, not knowing how many other substrates hepsin has in the kidney or in the urinary tract. The fact that this protease is expressed in other nephron segments seems to argue in favour of possible additional functions. Moreover, renal parameters may also be reflecting the consequences of the lack of hepsin in other organs. To properly study the renal consequences of the lack of hepsin-mediated processing of uromodulin we feel that a more appropriate and rigorous genetic approach would be needed, as for instance the generation of a knock-in mouse carrying mutation of uromodulin on the hepsin cleavage site. This is indeed a line of investigation that we plan to follow in the next years.

Although we believe that these considerations would undermine the significance of any study in *Hpn*^-/-^ mice, we anyway made considerable efforts to investigate possible renal phenotype in these mice.

We first evaluated renal parameters that were reported to be altered in *Umod* KO mice. Baseline levels of serum creatinine and blood urea nitrogen are similar in *Hpn* KO mice and wild-type littermates, suggesting normal renal function (see Discussion section, first paragraph). Also, genes shown to be differentially expressed in the kidneys of *Umod* KO mice (i.e. *Ptgs2*, *Slc12a3*, *Kcnj1*; Bachmann et al., Am J Physiol Ren Physiol 2005) are expressed at levels comparable with wild-type littermates in *Hpn* KO mice (see Figure 10).

Author response image 2.Expression level of renal genes in *Hpn*_-/-_ mice.Quantification by Real-time qPCR of the mRNA levels of cyclooxygenase 2 (Cox2) (*Ptgs2*), sodium-chloride co-transporter (NCC) (*Slc12a3*) and the renal outer medullary potassium channel (ROMK) (*Kcjn1*) in kidneys of *Hpn_-/-_*mice relative to age- and sex- matched wild-type controls (n = 3/group). Bars indicate average ratio ± s.e.m.**DOI:**
http://dx.doi.org/10.7554/eLife.08887.031

In the due time it was not possible to carry out thorough *in vivo* experiments to assess for instance if *Hpn* KO mice develop a phenotype upon different challenges (e.g. transurethral bacteria injection, aging-associated nephrocalcinosis, etc.).

We envisaged an alternative strategy by testing *ex vivo* anti-bacterial properties of urines from *Hpn* KO mice. This approach stems from the evidence that uromodulin can bind type 1-fimbriated uropathogenic *E. coli* through its high-mannose moieties (Pak et al., J Biol Chem 2001). Thanks to this property, it is thought that urinary uromodulin can compete with binding of type 1-fimbriated *E. coli* to high-mannose glycans of the urothelial membrane proteins uroplakins (Wu et al., PNAS 1996), thereby acting as a major urinary defence factor against bacterial infections. Although we still lack conclusive evidence of such a role of uromodulin from human studies, increased colonisation of the urinary bladder and more persistent bacteriuria in *Umod* KO mice upon bacteria injection (Bates et al., Kidney Int 2004; Mo et al., Am J Physiol Ren Physiol 2004) support this hypothesis.

We reasoned that testing the ability of mouse urines (or purified urinary proteins) to inhibit mannose-dependent adhesion of type 1-fimbriated *E. coli* could mimic the *in vivo* interactions expected to occur in experimental models of urinary tract infection.

For these experiments we employed 3 different assays testing *E. coli* adhesion: 1) agglutination of rat erythrocytes (Wu et al., PNAS 1996); 2) yeast agglutination (Mirelman et al., J Clin Microbiol 1980; Wang et al., BMC Microbiol 2012); 3) adhesion to the surface of urothelial cells (Orskov et al., Infect Immun1980; Martinez et al., EMBO J 2000). Haemagglutination and yeast agglutination assay were successfully set up to screen uropathogenic bacteria clinical isolates (in collaboration with Dr. Nicasio Mancini, Microbiology and Virology Laboratory, San Raffaele Hospital). Out of 16 *E. coli* isolates, we eventually selected 3 that clearly induced haemagglutination and mannose-sensitive yeast agglutination, confirming their phenotype as type 1-fimbriated (see Figure 11). The same isolates could also bind to the surface of urothelial cells in a mannose-sensitive manner.

Author response image 3.Yeast agglutination assays on uropathogentic *E. coli* clinical isolates.Representative images of yeast agglutination assay. The assay was performed incubating 10 µl of previously identified bacteria isolates (haemagglutination-positive isolates #9, 12, 10), grown in static condition (final concentration 2x10^9^ cells/ml), with 10 µl of *S. cerevisiae* yeast strain W303 Mata (final concentration of 10^9^ cells/ml). The appearance of white aggregates (arrows) indicates agglutination. UTI isolates were pre-incubated with D- mannose at a final concentration of 2.5% in order to identify mannose-sensitive agglutination, indicative of type 1-fimbriated phenotype. The assay was carried out in 3 replicates.**DOI:**
http://dx.doi.org/10.7554/eLife.08887.032

Unfortunately, these assays gave contrasting and inconclusive results when we employed human urine-purified uromodulin (tested range 0.1-1 µg/µl), mouse urinary proteins (0.2-0.7 µg/µl) or mouse urine (1:1-1:20 dilutions). A partial effect of purified uromodulin on yeast agglutination and bacteria adhesion to urothelial cells could be observed, while mouse urine and mouse urinary proteins had no effect in any assay. We observed some difference between wild-type and *Hpn*^-/-^ urinary protein samples in their inhibitory effect on bacterial adhesion to urothelial cells in a single experiment, but this result could not be convincingly reproduced.

Overall, we believe that such experiments would require extensive set up and analysis to be optimised for mouse urinary samples. As for experiments *in vivo* we feel that this would be well beyond the scope of the current work.

Nonetheless, we wish to point out that, for the considerations stated above, experiments in the *Hpn*^-/-^ model would eventually lead to results that could not be conclusively interpreted.